# Lunatic fringe-mediated Notch signaling regulates adult hippocampal neural stem cell maintenance

Fatih Semerci[1,2], William Tin-Shing Choi[1,2,3], Aleksandar Bajic[2,4], Aarohi Thakkar[2,4], Juan Manuel Encinas[2,5], Frederic Depreux[6], Neil Segil[7,8], Andrew K Groves[1,9,10], Mirjana Maletic-Savatic[1,2,4,10]*

[1]Program in Developmental Biology, Baylor College of Medicine, Houston, United States; [2]Jan and Dan Duncan Neurological Research Institute at Texas Children's Hospital, Houston, United States; [3]Medical Scientist Training Program, Baylor College of Medicine, Houston, United States; [4]Department of Pediatrics, Baylor College of Medicine, Houston, United States; [5]Achucarro Basque Center for Neuroscience and Ikerbasque, The Basque Science Foundation, Bizkaia, Spain; [6]Department of Cell Biology and Anatomy, Rosalind Franklin University of Medicine and Science, Chicago, United States; [7]Department of Stem Cell Biology and Regenerative Medicine, Keck School of Medicine, University of Southern California, Los Angeles, United States; [8]Caruso Department of Otolaryngology, Keck School of Medicine, University of Southern California, Los Angeles, United States; [9]Department of Molecular and Human Genetics, Baylor College of Medicine, Houston, United States; [10]Department of Neuroscience, Baylor College of Medicine, Houston, United States

*For correspondence: maletics@bcm.edu

Competing interests: The authors declare that no competing interests exist.

**Abstract** Hippocampal neural stem cells (NSCs) integrate inputs from multiple sources to balance quiescence and activation. Notch signaling plays a key role during this process. Here, we report that Lunatic fringe (*Lfng*), a key modifier of the Notch receptor, is selectively expressed in NSCs. Further, Lfng in NSCs and Notch ligands Delta1 and Jagged1, expressed by their progeny, together influence NSC recruitment, cell cycle duration, and terminal fate. We propose a new model in which Lfng-mediated Notch signaling enables direct communication between a NSC and its descendants, so that progeny can send feedback signals to the 'mother' cell to modify its cell cycle status. Lfng-mediated Notch signaling appears to be a key factor governing NSC quiescence, division, and fate.

## Introduction

The ability of the hippocampal neurogenic niche to respond to ever-changing stimuli throughout the lifespan requires that it carefully maintain its finite stock of neural stem cells (NSCs) (*Kuhn et al., 2005*, *1996*). NSCs are the primary stem cells of the niche and are quite plastic in their responses: for example, social isolation seems to increase NSC self-renewal (*Dranovsky et al., 2011*), whereas running stimulates neurogenesis (*van Praag et al., 1999*), and seizures prompt NSCs to transform directly into reactive astrocytes (*Sierra et al., 2015*). Taking into account that nearly 80% of the newborn progeny of NSCs die by apoptosis (*Sierra et al., 2010*), the neurogenic niche needs to ensure not only that it responds properly to stimuli but also that NSCs, once they are activated, are cycling enough to produce sufficient progeny.

One way to achieve this optimization would be for the progeny to somehow communicate their status back to the ancestor NSC. Communication between 'mother' and 'daughter' cells is known to regulate lateral inhibition in the vertebral neural tube (*Nikolaou et al., 2009*) and *Drosophila* oocytes (*Zhao et al., 2000*), and it occurs during angiogenesis (*Benedito et al., 2009*) and oncogenesis (*Xu et al., 2012*); in each case this communication involves Notch signaling (*Haines and Irvine, 2003*; *LeBon et al., 2014*; *Stanley and Okajima, 2010*; *Taylor et al., 2014*; *Yang et al., 2005*). Notch signaling is evolutionarily conserved (*Andersson et al., 2011*) and plays a key role in development through diverse effects on differentiation, proliferation, and survival (*Alunni et al., 2013*; *Breunig et al., 2007*; *Giachino and Taylor, 2014*) that depend on signal strength (*Basch et al., 2016*; *Chapouton et al., 2010*; *Gama-Norton et al., 2015*; *Ninov et al., 2012*; *Shimojo et al., 2008*) and cellular context (*Basak et al., 2012*; *Farnsworth et al., 2015*; *Lugert et al., 2010*). In the fetal brain, Notch activity maintains embryonic NSCs in an undifferentiated state (*Louvi and Artavanis-Tsakonas, 2006*) by suppressing pro-neural gene expression (*Gaiano et al., 2000*; *Ishibashi et al., 1994*; *Lütolf et al., 2002*) and supporting progenitor survival (*Androutsellis-Theotokis et al., 2006*; *Louvi and Artavanis-Tsakonas, 2006*). In the adult brain, Notch seems to influence quiescence, cycling, and exit of neuroprogenitors from the cell cycle, acting most likely in a cell-autonomous fashion (*Ables et al., 2010*; *Basak et al., 2012*; *Breunig et al., 2007*; *Ehm et al., 2010*; *Ehret et al., 2015*). Despite considerable advances in our understanding of Notch signaling, however, we do not know the precise cell-specific mechanism that might connect hippocampal NSCs and their progeny.

We hypothesized that, if Notch does facilitate communication between the mother NSC and its daughter cells, it might do so through the fringe proteins (Lunatic, Manic, Radical), which are known regulators of Notch signaling. Glycosylation of Notch receptors by fringe proteins affects the intracellular cleavage of the heterodimeric receptor complex and generation of the Notch1 Intra Cellular Domain (NICD) following ligand binding. Typically, NICD production increases upon binding by Delta-like (Dll) and decreases following Jagged1 (Jag1) binding (*LeBon et al., 2014*; *Stanley and Okajima, 2010*; *Taylor et al., 2014*; *Yang et al., 2005*); differential Notch cleavage ensures varying expression of downstream cell cycle genes (*Chapouton et al., 2010*; *Isomura and Kageyama, 2014*; *Nellemann et al., 2001*; *Ninov et al., 2012*; *Yoshiura et al., 2007*). To examine whether fringe proteins are present in NSCs, we systematically queried existing expression databases, such as the Allen Brain Atlas (*Lein et al., 2007*) and GENSAT (*Gong et al., 2003*), and discovered that Lunatic fringe (*Lfng*) appears to be selectively expressed in adult hippocampal NSCs. We confirmed this observation using *Lfng*-eGFP mice and further demonstrated the potency of *Lfng*-eGFP-expressing cells using a newly-generated *Lfng*-CreER$^{T2}$ line for lineage tracing. The selective expression of *Lfng* in NSCs has enabled us to explicitly examine the role of Notch signaling in NSC regulation. Here, using several new transgenic mouse models, we unveil a novel Notch-based mechanism that mediates direct communication between NSCs and their progeny to control NSC quiescence and activation.

## Results

### *Lfng*-eGFP reporter mice specifically mark NSCs of the dentate gyrus

Our database search indicated that *Lfng* might selectively label hippocampal NSCs, prompting us to fully characterize the *Lfng*-eGFP transgenic mouse. Spatially, *Lfng*-eGFP expression in the hippocampus was restricted to the subgranular zone (SGZ) of the dentate gyrus, where neural stem and progenitor cells reside (*Seri et al., 2001*) (*Figure 1A*, left panel). *Lfng*-eGFP$^+$ cells closely resembled NSCs: their triangular soma was located in the SGZ, from which a single radial process extended orthogonally and spanned the granule cell layer, ending in fine arborizations within the molecular layer (*Figure 1A*, right panel). In situ hybridization for Lfng mRNA confirmed that eGFP expression recapitulated the endogenous Lfng expression pattern in the adult dentate gyrus (*Figure 1B*). To confirm that eGFP expression accurately reflects *Lfng* expression, we crossed *Lfng*-eGFP with *Lfng*$^{Tm1Grid}$ mice that carry beta-galactosidase (β-Gal) insertion in the *Lfng* locus (*Zhang and Gridley, 1998*). In the resulting *Lfng*$^{Tm1Grid}$; *Lfng*-eGFP mice, β-Gal and eGFP co-localized (*Figure 1C*), demonstrating that the regulatory elements driving eGFP expression in *Lfng*-eGFP mice are active in the same cells that expressed β-Gal in *Lfng*$^{Tm1Grid}$ mice. Finally, Nestin, Sox2, Vimentin, and GFAP,

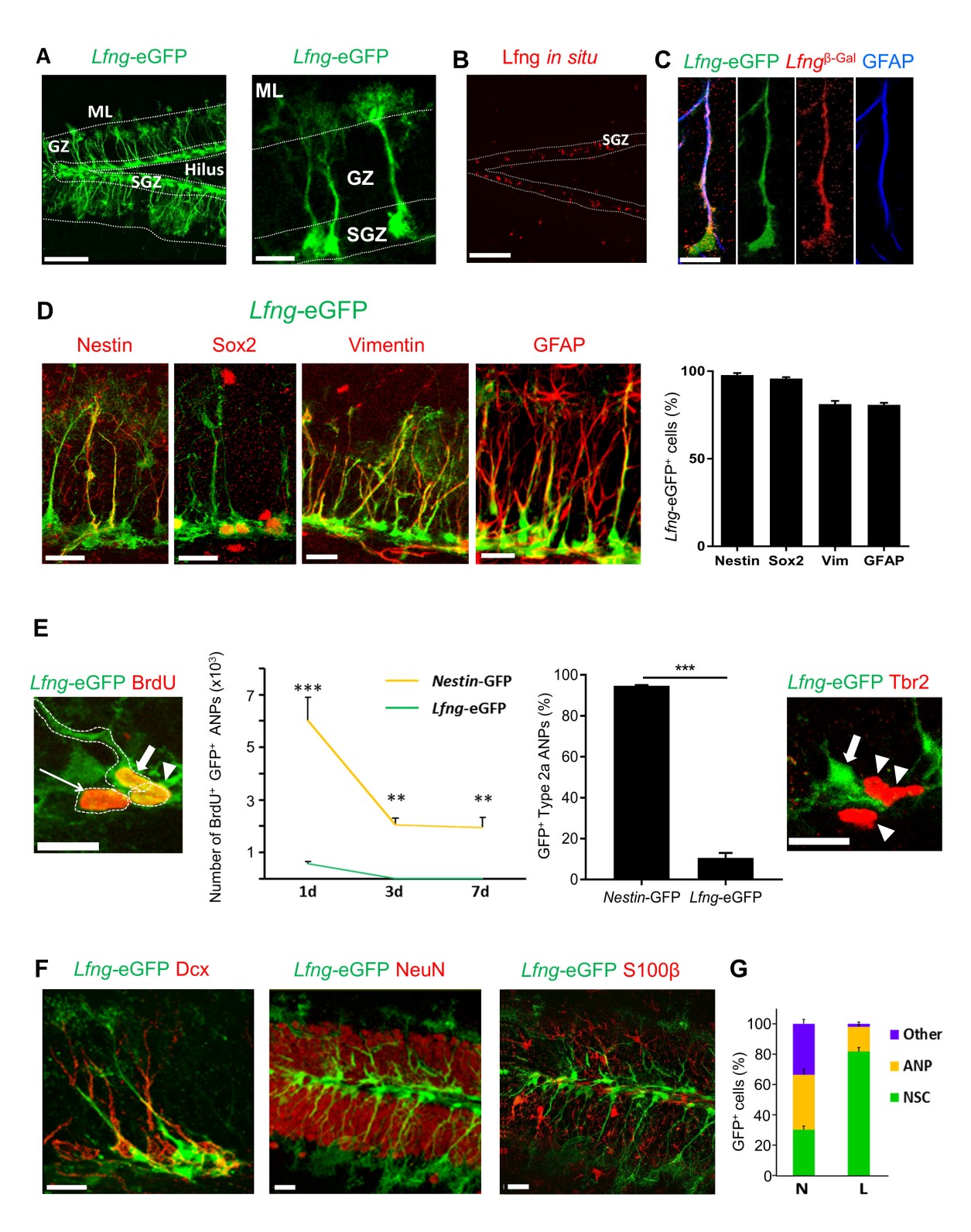

**Figure 1.** *Lfng*-eGFP is expressed in the NSCs of dentate gyrus. (**A**) *Left panel:* Confocal photomicrograph of the dentate gyrus in 2 month-old *Lfng*-eGFP mouse shows *Lfng*-eGFP expressing cells in the subgranular zone only. *Right panel: Lfng*-eGFP expressing cells have typical NSC morphology: triangular cell body in the subgranular zone, a single radial process spanning the granular zone, and fine terminal arborizations in the molecular layer. ML=molecular layer, GZ=granule cell zone, SGZ=subgranular zone. (**B**) In situ hybridization against the Lfng mRNA shows probe expression in the SGZ.

*Figure 1 continued on next page*

Figure 1 continued

Dotted lines indicate the borders of SGZ. (C) In a double transgenic $Lfng^{Tm1Grid}$;$Lfng$-eGFP mouse, β-galactosidase (β–Gal) and eGFP are co-expressed, confirming that $Lfng$ promoter guiding the eGFP expression is active in the same cells that express β–Gal. $Lfng^{Tm1Grid}$=$Lfng^{β-Gal}$ mouse that carries β-Gal insertion in the $Lfng$ locus. (D) $Lfng$-eGFP colocalizes with other NSC markers, such as Nestin, Sox2, Vimentin, and GFAP. Confocal photomicrographs of the representative examples and relative quantitation is shown. (E) eGFP is briefly retained in the first progeny of NSC. *Left panel:* $Lfng$-eGFP expressing cell (thick arrow) and its immediate ANP progeny (arrowhead) are both BrdU$^+$ and in cytoplasmic contact. BrdU$^+$ ANP not in the cytoplasmic contact with the BrdU$^+$ NSC has very low eGFP expression (thin arrow). *Left graph:* eGFP is cleared from BrdU$^+$ ANPs between 1 to 3 days following a single dose of BrdU, corresponding to the half-life of GFP protein (N = 3–5 per group, p<0.001 or p<0.0001 for pairwise comparisons of all timepoints, ANOVA with Tukey post-hoc test). *Right graph:* $Lfng$-eGFP is expressed only in minority of Type2a ANPs (Sox2$^+$ GFAP$^-$ cells in the SGZ; 10.58 ± 2.38, N = 3), unlike *Nestin*-GFP (94.7 ± 0.34, N = 4). *Right panel:* Late, Tbr2$^+$ ANPs (arrowheads) do not express $Lfng$-eGFP. Arrow points to $Lfng$-eGFP NSC. (F) $Lfng$-eGFP does not co-localize with the markers of neuronal lineage (Dcx$^+$ neuroblasts and immature neurons, and NeuN$^+$ granule cells) nor S100β$^+$ astrocytes. (G) While in the *Nestin*-GFP (N) mice GFP is expressed in NSCs, ANPs, and other cell types throughout the dentate gyrus in approximately equal proportions (30.16 ± 2.43 NSCs, 36.12 ± 3.73 ANPs, 33.71 ± 2.81 other cell types), in the $Lfng$-eGFP (L) mice it is expressed predominantly in NSCs (81.68 ± 2.62% NSCs, 16.37 ± 1.54% ANPs, 1.95 ± 1.14% other cell types; N = 4 per genotype). Bars represent mean±SEM. **p<0.001, ***p<0.0001. *Scale bars* = 100 μm (A left, (B), 20 μm (A right, (C–F). See *Figure 1—figure supplement 1* for further details.

The following figure supplement is available for figure 1:

**Figure supplement 1.** $Lfng$-eGFP labels a small number of non-NSC cell types in the dentate gyrus.

known to label NSCs, were all expressed in $Lfng$-eGFP$^+$ cells (*Figure 1D*). Together, these diverse lines of evidence indicate that $Lfng$-eGFP is a marker of NSCs.

A small proportion of the eGFP-expressing cells in the SGZ had a round amplifying neuroprogenitors (ANP) -like morphology with no associated processes. To determine whether ANPs express eGFP, we crossed $Lfng$-eGFP with the *Nestin*-CFP$^{nuc}$ mice, which express CFP in the nuclei of both NSCs and ANPs (*Encinas et al., 2006*). In the double transgenic mice, all $Lfng$-eGFP$^+$ cells co-expressed CFP (*Figure 1—figure supplement 1A*), but only 7.4 ± 1.9% (mean ± SEM, N = 3) of the CFP-labeled ANPs contained eGFP. The eGFP$^+$ ANPs were in close proximity to dividing BrdU$^+$ NSCs and in all cases were in cytoplasmic connection with the mother NSC (*Figure 1E*, left panel), which suggests that the asymmetric division had not yet finished. ANPs not cytoplasmically connected with the NSCs had little if any eGFP (*Figure 1E*, left panel).

We next examined how long it took newborn ANPs to clear eGFP, using BrdU pulse-and-chase labeling. eGFP was cleared from $Lfng$-eGFP$^+$ ANPs, but not from *Nestin*-GFP$^+$ (*Mignone et al., 2004*) ANPs, between days 1 and 3 (*Figure 1E*, left graph). The clearance correlates with the half-life of GFP (*Corish and Tyler-Smith, 1999*), suggesting that the eGFP in ANPs is likely from the protein retained by the first progeny of $Lfng$-eGFP$^+$ NSCs. Indeed, only 10% of Type 2a cells (Sox2$^+$ early ANPs) were eGFP$^+$ in $Lfng$-eGFP mice compared to almost 95% in *Nestin*-GFP mice (*Figure 1E*, right graph). No Type 2b cells (late ANPs) expressed $Lfng$-eGFP (*Figure 1E*, right panel). Together, these data strongly suggest that most of the $Lfng$-eGFP$^+$ ANPs are the immediate progeny of NSCs that retained some eGFP due to sequestration of the protein during cell division. ANPs quickly lose eGFP, implying that $Lfng$-eGFP expression is selective for NSCs and that $Lfng$ is active in NSCs but not in ANPs.

$Lfng$-eGFP was not detected in neuroblasts/immature neurons, granule cells, or astrocytes (*Figure 1F*). In a minority of cases (0.8%) we observed the endothelial cell marker CD31 overlapping with $Lfng$-eGFP (*Figure 1—figure supplement 1B*). This is not surprising, as endothelial tip cells express Lfng during tip-stalk cell selection after exposure to angiogenic factors (*Benedito et al., 2009*). Further, we observed a few $Lfng$-eGFP$^+$ cells, some co-labeled with GFAP$^+$ or S100β$^+$, randomly distributed in the dentate gyrus (0.9%; *Figure 1—figure supplement 1C*). The overall quantification of the cell-type expression of eGFP indicated that the majority of eGFP$^+$ cells in $Lfng$-eGFP mice were NSCs (81.7 ± 2.6%), while only 16.4 ± 1.5% were ANPs and 1.9 ± 1.1% were other cell types (*Figure 1G*). In contrast, in *Nestin*-GFP mice, the majority of GFP$^+$ cells were ANPs (36.1 ± 3.7%), followed by other cell types (33.7 ± 2.8%) and NSCs (30.2 ± 2.4%). These data emphasize the specificity of $Lfng$-eGFP expression for adult hippocampal NSCs, compared to the low selectivity of *Nestin*-GFP expression for NSCs.

At any given time, the majority of NSCs is quiescent; only about 1–4% of them divide under physiological conditions (*Encinas et al., 2011*; *Knobloch et al., 2014*). This property enables them to

escape treatment with cytostatic drugs, such as temozolomide (*Knobloch et al., 2014*). Indeed, only 15.0 ± 3.0% of the eGFP$^+$ cells were lost in *Lfng*-eGFP mice following temozolomide treatment (N = 4, p=0.27), compared to 35.0 ± 4.0% in the *Nestin*-GFP mice (N = 4, p=0.0035). This suggests that the *Lfng*-eGFP$^+$ population consists mostly of quiescent NSCs, which we confirmed by independent quantification of NSCs and ANPs (*Figure 2A*). Further, if *Lfng*-eGFP$^+$ cells are true NSCs, then both physical exercise and electroconvulsive shock (ECS) should activate them (*Lugert et al., 2010*; *Madsen et al., 2000*; *Segi-Nishida et al., 2008*; *Suh et al., 2007*; *van Praag et al., 1999*, *2005*; *Warner-Schmidt and Duman, 2006*). We therefore tested whether *Lfng*-eGFP$^+$ NSCs respond to these stimuli. Mice were either administered a single ECS for four consecutive days or subjected to voluntary access to a running wheel for a week, followed by a single injection of BrdU. Sham controls either did not receive ECS or had a locked running wheel in their cage. Mice were sacrificed 24 hr after the BrdU injection and the BrdU$^+$ NSCs were quantified. A robust increase in the percentage of BrdU$^+$ *Lfng*-eGFP$^+$ NSCs in both conditions (N = 4 per group in ECS, p=0.0006; and N = 6 per group in PE, p=0.0271) indicated that *Lfng*-eGFP$^+$ NSCs are plastic, as expected (*Figure 2B*).

We next assessed the net outcome of neurogenesis using BrdU pulse-and-chase labeling and compared *Lfng*-eGFP and *Nestin*-GFP mice to determine whether there were any gross differences in these two different transgenic lines. Over the course of 30 days, the total number of BrdU$^+$ cells in the dentate gyrus followed the same pattern of decline (R = 0.9992; *Figure 2—figure supplement 1A*). The decay was rapid between 1 and 3 days post-BrdU injection (dpi; 100–105 BrdU$^+$ cells lost per hour), slowed to 31–27 BrdU$^+$ cells/hr between 3 and 7dpi, and reached a stable rate of 3–4 BrdU$^+$ cells/hr between 7-15dpi, and 2–3 cells/hr between 15 and 30dpi. Neurogenesis in *Lfng*-eGFP mice thus follows the same survival pattern observed in *Nestin*-GFP mice (*Kempermann et al., 2004*; *Kronenberg et al., 2003*; *Sierra et al., 2010*). Further, the *Lfng*-eGFP$^+$ NSCs produced the same neurogenic progeny as *Nestin*-GFP$^+$ NSCs over 30 days, providing further assurance that there are no gross differences in these two transgenic lines (*Figure 2C*). This is important, because it establishes the benchmark validity of the *Lfng*-eGFP mice for studies of neurogenesis.

Finally, adult neurogenesis declines with age (*Kuhn et al., 1996*), at least in part because of a decline in the NSC population (*Encinas et al., 2011*). Thus, if *Lfng*-eGFP$^+$ cells are NSCs, then their number should decrease over time. This is, in fact, what we observed; over an 18 month period the number of NSCs (*Figure 2D*, left panel), the number of BrdU$^+$ cells (*Figure 2—figure supplement 1B*, left panel) and the proportion of BrdU$^+$ NSCs (*Figure 2—figure supplement 1B*, right panel) were similar between the *Lfng*-eGFP and *Nestin*-GFP mice. Unlike the *Nestin*-GFP mice, however, the specificity of *Lfng*-eGFP expression for NSCs did not change over time (*Figure 2D*, right panel; *Figure 2—figure supplement 1C*). This is a crucial finding because it signifies the utility of the *Lfng*-eGFP mouse model for specific studies of NSC biology during aging.

## Lfng-expressing NSCs generate diverse progeny

The claim that *Lfng*-eGFP$^+$ cells are functional NSCs capable of producing diverse progeny requires lineage tracing. Thus, we generated a *Lfng*-CreER$^{T2}$ transgenic line using the same bacterial artificial chromosome used to generate *Lfng*-eGFP mice, modified by inserting CreER$^{T2}$ in front of the transcription start site of the *Lfng* (RP23-270N2; *Figure 3A*). To verify that CreER$^{T2}$ is selectively expressed in NSCs, we bred *Lfng*-CreER$^{T2}$ mice to the AI14 (RCL-tdT) reporter line (Jackson Lab Stock no: 007908) (*Madisen et al., 2010*) and then to *Lfng*-eGFP mice. One day following induction, 89.1 ± 4.0% of tdTomato$^+$ cells were eGFP$^+$ and 37.9 ± 1.5% of *Lfng*-eGFP$^+$ NSCs were tdTomato$^+$ (*Figure 3B*). No tdTomato expression was observed after vehicle administration (*Figure 3—figure supplement 1A*).

To further verify that Lfng-expressing NSCs are dividing, we crossed the *Lfng*-CreER$^{T2}$ mice with the *iDTR* mice, in which the diphtheria toxin receptor (DTR, a.k.a. Hbegf, simian Heparin-binding epidermal growth factor-like growth factor) is conditionally expressed under the control of Cre-activated Rosa26 locus (*Buch et al., 2005*). Activation of this receptor by diphtheria toxin selectively kills DTR-expressing cells (*Buch et al., 2005*). Fifteen days following induction of DTR in *Lfng*-CreER$^{T2}$; *iDTR* mice and activation by diphtheria toxin, we observed a significant reduction in both NSCs (36.8 ± 1.5%; N = 3–4 per group; p=0.0244) and the Ki67$^+$ cells (57.3 ± 2.4%; p<0.0001) (*Figure 3—figure supplement 1B*). As neither DTR expression nor the high dose of diphtheria toxin alone cause cell death (*Arruda-Carvalho et al., 2011*; *Buch et al., 2005*; *Gropp et al., 2005*), our data confirm the

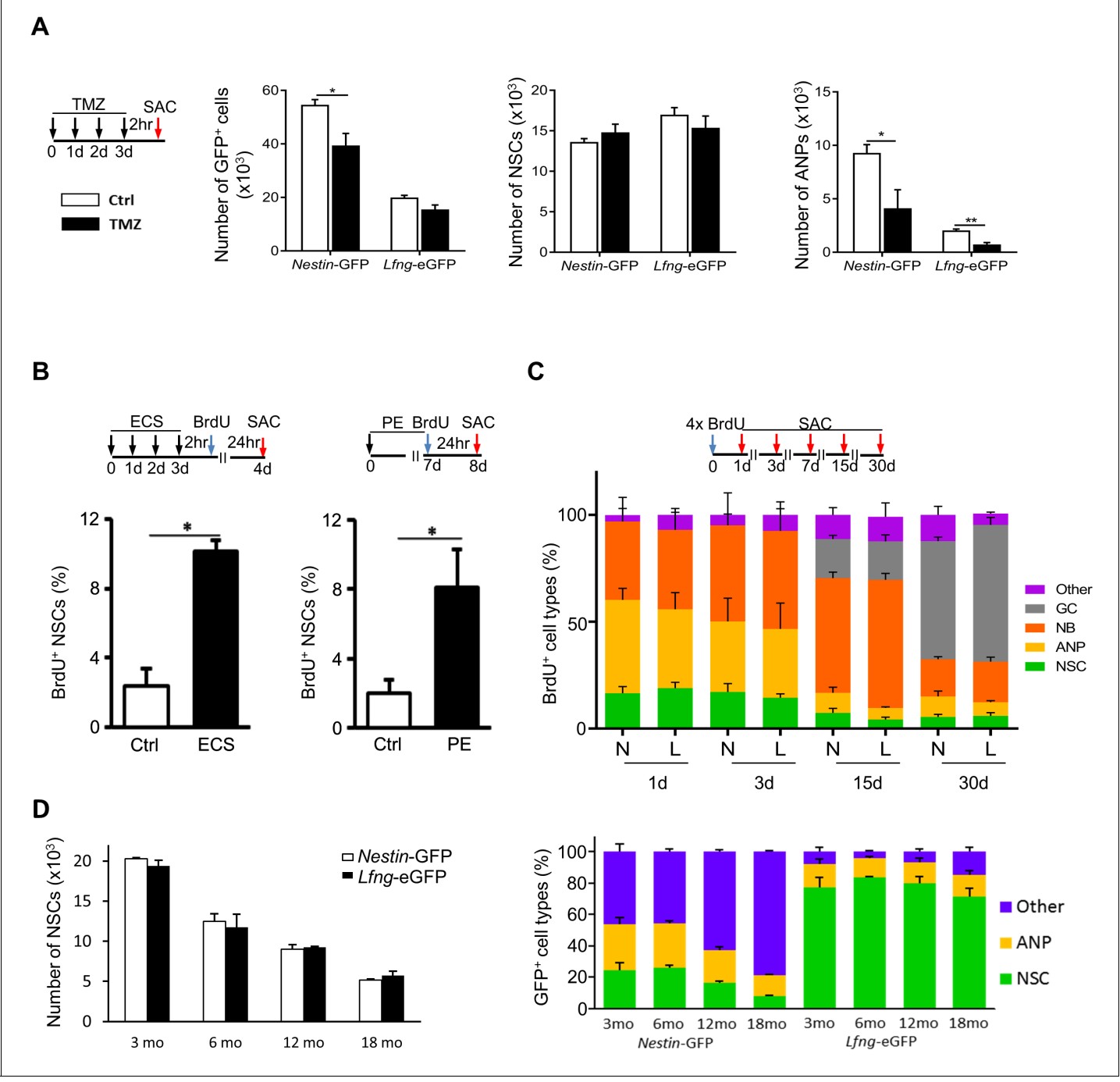

**Figure 2.** *Lfng*-eGFP-expressing cells are functional NSCs. (**A**) Most *Lfng*-eGFP$^+$ NSCs are quiescent. Bar graphs represent the total number of GFP$^+$ cells (left panel), GFP$^+$ NSCs (middle panel) and GFP$^+$ ANPs (right panel) in 3 month-old *Nestin*-GFP and *Lfng*-eGFP mice (N = 4 per genotype) treated with temozolomide (TMZ). The difference between the two mouse models is most notable with respect to ANPs: while *Nestin*-GFP labels a large number of ANPs, *Lfng*-eGFP does not - it labels primarily quiescent cells. (**B**) Electroconvulsive shock (ECS) and physical exercise (PE) both activate *Lfng*-eGFP$^+$ NSCs (N = 4 per group in ECS and N = 6 per group in PE). (**C**) *Lfng*-eGFP NSCs produce neuronal progeny. The relative number of newborn, BrdU$^+$ progeny was quantified over a 30 day period (NSCs: *Nestin*-GFP$^+$ or *Lfng*-eGFP$^+$ cells with GFAP$^+$ radial processes; ANPs: GFAP$^-$ Dcx$^-$ NeuN$^-$; NBs: Dcx$^+$ neuroblasts and immature neurons; GCs: NeuN$^+$; Other: BrdU$^+$ Dcx$^-$ NeuN$^-$ cells outside the SGZ; N = 4 per genotype per timepoint). Cumulative BrdU paradigm (four 150 mg/kg injections given 2 hr apart) was used to increase the yield of labeled newborn cells. N=*Nestin*-GFP mice, L=*Lfng*-eGFP mice. (**D**) The number of *Lfng*-eGFP$^+$ NSCs declines over an 18 month period comparably to the number of *Nestin*-GFP$^+$ NSCs (left panel; N = 4 per timepoint per genotype). However, the contribution of GFP$^+$ cell types in the *Nestin*-GFP and *Lfng*-eGFP mice differs at different age (right panel). While *Lfng*-eGFP remains selective for NSCs during aging (p>0.15 for all timepoints, Tukey post-hoc test), *Nestin*-GFP labels

*Figure 2 continued on next page*

*Figure 2 continued*

significantly more non-neuroprogenitors in older mice (p<0.002; Tukey post-hoc test). Bars represent mean±SEM. NS=non-significant, *p<0.05, **p<0.001. See *Figure 2—figure supplement 1* for further details.

The following figure supplement is available for figure 2:

**Figure supplement 1.** The specificity of *Lfng*-eGFP expression for NSCs does not change over time.

proliferative properties of Lfng-expressing NSCs and indicate that even partial elimination of NSCs leads to a dramatic decrease of cycling cells in the SGZ neurogenic niche.

Next, we examined the lineage of tdTomato$^+$ NSCs in *Lfng*-CreER$^{T2}$;RCL-tdT mice (*Figure 3C–E* and *Figure 3—figure supplement 1C,D*). We first focused on NSCs and their immediate progeny. Within 1–2 days after a single dose of tamoxifen (200 mg/kg), induced NSCs with radial processes and fine arborizations in the molecular layer appeared as tdTomato$^+$ cells. They expressed Sox2 and GFAP, further supporting recombination in NSCs (*Figure 3C*, left panel). They divided both asymmetrically, to give rise to ANPs (*Figure 3C*, middle panel) and symmetrically, to give rise to two Sox2$^+$ cells connected via cytoplasm and both having prominent GFAP$^+$ radial processes (*Figure 3C*, right panel). We then performed fate-mapping for two months, using a lower dose (120 mg/kg) of tamoxifen to sparsely induce cells and allow visualization of progeny morphology. Within 3 days of induction we detected tdTomato$^+$ Tbr2$^+$ late ANPs (Type 2b cells), followed by Dcx$^+$ immature neurons 7 days after (*Figure 3D*). Thirty days following induction, tdTomato$^+$ cells expressed the granule cell marker NeuN (*Figure 3D*) and had extended processes reaching the molecular layer, prominent dendritic spines, and an axon extending throughout the hilus to the CA3 region (*Figure 3—figure supplement 1C*). These data demonstrate that Lfng-expressing cells generate neurons and enable detailed studies of the neuronal lineage linked directly to a single NSC.

In addition to neuronal progeny, we also detected tdTomato$^+$ S100$\beta^+$ astrocytes in the granule cell layer (*Figure 3E*, left panel) and tdTomato$^+$ Sox2$^+$ GFAP$^+$ astrocyte-like cells in the hilus (*Figure 3E*, right panel). These may represent either terminally differentiated NSCs or the progeny of *Lfng*-expressing NSCs; characterization of these cells is beyond the scope of this paper and will await future studies. Finally, as in the *Lfng*-eGFP mice, we found tdTomato$^+$ tip cells in *Lfng*-CreER$^{T2}$; RCL-tdT mice (*Figure 3—figure supplement 1D*), further supporting the finding that the same regulatory elements control the expression of both eGFP and CreER$^{T2}$ in these two different mouse models.

Quantitative evaluation of the lineage of Lfng-expressing NSCs provided additional information on the progeny and differentiation timeline (*Figure 3F*). Initially, most of the population consisted of NSCs with some ANPs and astrocyte-like cells (GFAP$^+$, S100$\beta^-$). Over the course of 60 days, the contribution of NSCs to the population of tdTomato$^+$ cells diminished because of the growing contribution of ANPs, immature neurons and granule cells. Interestingly, the population of ANPs and immature neurons remained stable from 15 days on, indicating a continuous replenishment of these cell types. The granule cell number steadily increased from 15 days on and represented about 30% of all tdTomato$^+$ cells at 60 days following induction. Surprisingly, the number of astrocyte-like cells did not change over the course of 60 days, although there was one change: these cells were mostly GFAP$^+$, S100$\beta^-$ between 1 and 15 days and mostly GFAP$^+$, S100$\beta^+$ thereafter. It may be that there is a steady conversion of NSCs into astrocyte-like cells that is independent of neurogenesis. Finally, we occasionally detected other tdTomato$^+$ cell types (mostly tip cells and their descendant endothelial cells), which constituted 6.8 ± 1.8% of the population at 60 days. In summary, these data demonstrate that Lfng-expressing cells in the SGZ are NSCs, able to self-renew and produce both neuronal (majority) and astrocytic (minority) lineage in the adult dentate gyrus. Collectively, these data establish *Lfng*-eGFP and *Lfng*-CreER$^{T2}$ as novel mouse models enhancing the existing repertoire of tools (*Semerci and Maletic-Savatic, 2016*). With these new mouse models, the field will be able to study the intrinsic properties of NSCs distinct from their progeny and gain a deeper understanding of the mechanisms of quiescence and lineage potential in the healthy brain, during aging, and in disease.

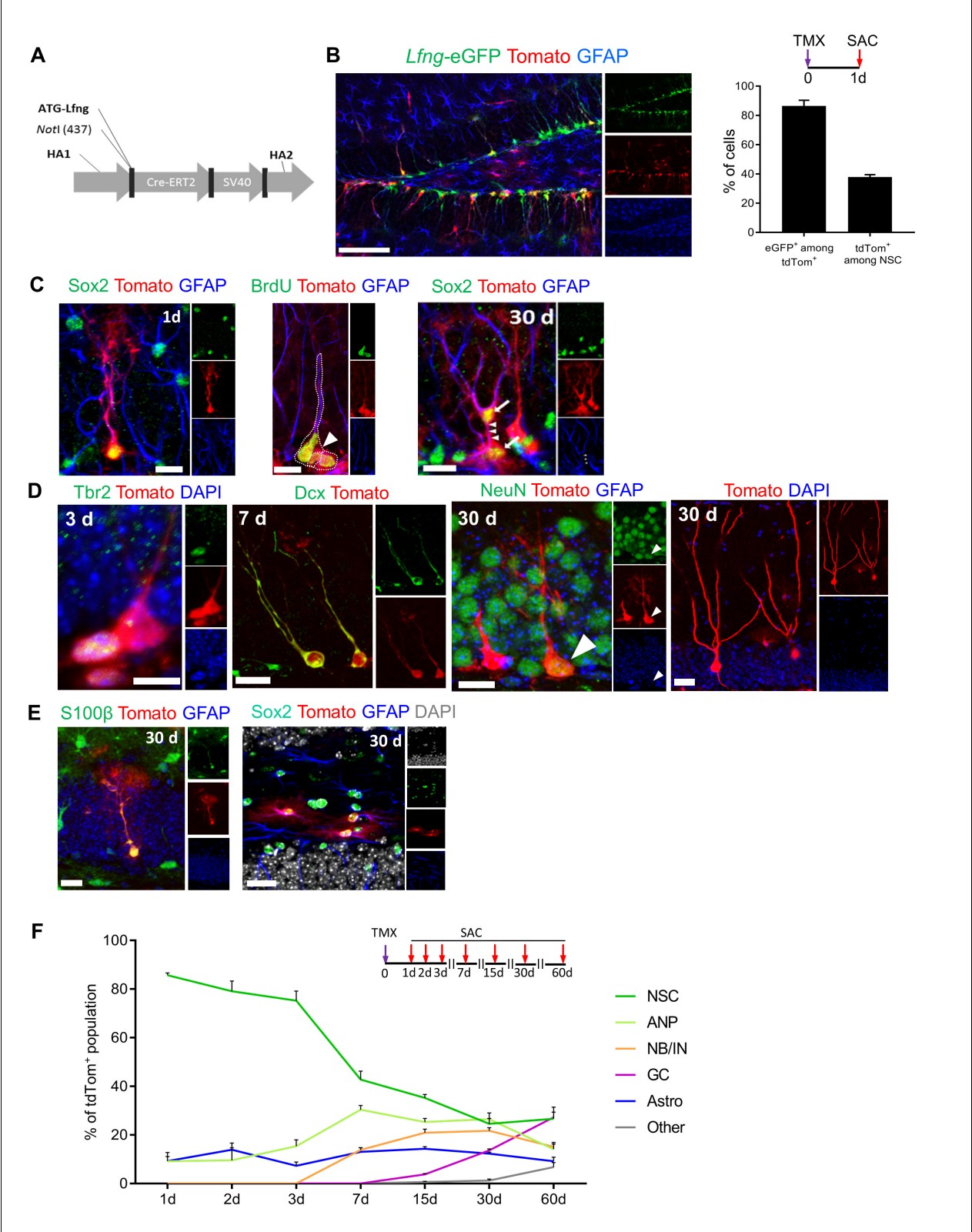

**Figure 3.** Lfng-expressing NSCs generate diverse progeny. (**A**) The map of bacterial artificial chromosome (BAC) construct used to generate the *Lfng*-CreER[T2] mouse. (**B**) In *Lfng*-CreER[T2]; RCL-tdT; *Lfng*-eGFP triple transgenic mouse, tdTomato[+] and eGFP[+] co-expressing cells demonstrate the specificity of the *Lfng*-CreER[T2] line to NSCs. *Left panel:* Confocal photomicrograph of the dentate gyrus of a 6 month-old mouse shows the overlapping expression of eGFP and CreER[T2]-controlled tdTomato one day following tamoxifen injection (TMX; 120 mg/kg). *Right panel:* Quantification of the co-
*Figure 3 continued on next page*

*Figure 3 continued*

expression of tdTomato[+] and eGFP[+] in induced *Lfng*-CreER[T2]; RCL-tdT mice (N = 3). Bars represent mean±SEM. (**C**) TMX-induced cells have NSC morphology and divide both asymmetrically and symmetrically. *Left panel*: tdTomato[+] NSC co-expresses GFAP and Sox2. *Middle panel*: tdTomato[+] NSC (arrow) divides asymmetrically to produce ANP (arrowhead). *Right panel*: NSC divides symmetrically to produce two cells (arrows) with prominent GFAP[+] radial processes. *Scale bars = 20 µm.* (**D**) tdTomato[+] NSCs produce new neurons through established cascade of cell types, from Tbr2[+] late ANPs (Type 2b cells), through Dcx[+] immature neurons, to NeuN[+] granule cells. *Scale bars = 20 µm.* (**E**) tdTomato[+] NSCs also produce S100*β* astrocytes (or astrocyte-like cells) within the granule cell layer (*left panel*), as well as stellar Sox2[+] GFAP[+] cells in the hilus (*right panel*) *Scale bars = 20 µm.* (**F**) Fate mapping of tdTomato[+] cells following TMX induction in *Lfng*-CreER[T2]; RCL-tdT mice reveals that NSCs form the majority (85.7% ± 0.9) of the induced cells at 1dpi, but progressively decline in ratio as they give rise to different progeny over the course of 2 months (N = 3–5 per timepoint). See *Figure 3—figure supplement 1* for further details.

The following figure supplement is available for figure 3:

**Figure supplement 1.** *Lfng*-CreER[T2] expressing cells are NSCs giving rise to newborn neurons.

## Notch pathway elements are present in the SGZ NSC niche

The selective expression of *Lfng* in adult SGZ NSCs raises the question about its functional role in these cells. Surprisingly, the biological role of *Lfng* in adult SGZ NSCs has never been examined, despite its known function as a direct transcriptional target of Notch (*Morales et al., 2002*) and the importance of Notch signaling for adult NSC maintenance (*Ables et al., 2010*; *Breunig et al., 2007*; *Ehm et al., 2010*; *Giachino and Taylor, 2014*). Lfng N-glycosylates Notch receptors (*Moloney et al., 2000*), affecting the intracellular cleavage of the heterodimeric receptor complex and generation of the NICD following ligand binding: typically, Lfng modification of Notch elevates NICD production upon Dll1 binding, but decreases it following engagement to Jag1 (*Haines and Irvine, 2003*; *LeBon et al., 2014*; *Stanley and Okajima, 2010*; *Taylor et al., 2014*; *Yang et al., 2005*). This might lead to differential activation of cell cycle genes downstream of Notch, depending on which type of ligand dominates (*Chapouton et al., 2010*; *Isomura and Kageyama, 2014*; *Nellemann et al., 2001*; *Ninov et al., 2012*; *Shimojo et al., 2008*; *Yoshiura et al., 2007*).

In agreement with published data (*Breunig et al., 2007*; *Lavado et al., 2010*), we found the key elements of the Notch pathway expressed in the NSCs and surrounding progeny: Notch1 on the surface of NSCs, Jag1 on presumed ANPs surrounding NSCs, and Dll1 on granule neurons neighboring the apical parts of the NSC soma (*Figure 4A*). In addition, Hes5 (an indicator of canonical Notch pathway; *Imayoshi et al., 2010*; *Lugert et al., 2010*) was present in some but not all *Lfng*-eGFP[+] NSCs (*Figure 4A*).

We further examined Notch signaling in NSCs using CBF:H2B-Venus (JAX 020942) (*Duncan et al., 2005*), a Notch reporter mouse in which YFP variant Venus is expressed under the control of CBF1 promoter. Approximately half of the NSCs had active Venus[+] Notch signaling at any given time, whereas Venus expression declined in early (Sox2[+]) and late (Tbr2[+]) ANPs as well as in neuroblasts and immature neurons (Dcx[+]) (*Figure 4B*). Notch signaling was turned on again in mature granule neurons as most of the NeuN[+] cells were positive for Venus signal with various degrees of intensity (*Figure 4B*), in agreement with the previous reports (*Brandt et al., 2010*; *Breunig et al., 2007*).

Although CBF:H2B-Venus mice are a good tool to evaluate the presence or absence of Notch signaling, the accumulation of Venus fluorescence protein and high fluorescence intensity in the GZ (*Figure 4B*) make it hard to perform reliable intensity quantifications in the SGZ where NSCs reside. We therefore turned to NICD1 as a more reliable measure of Notch signaling intensity. To verify the reliability of NICD1 staining, we reasoned that NICD1 should be present in cells expressing Venus. Indeed, the NICD1 expression overlapped with Venus expression in CBF:H2B-Venus mice: over 90% in SGZ and over 95% in GZ (*Figure 4C*).

These expression data provide the foundation for our hypothesis that Notch signaling, mediated by Lfng, represents a mode of communication between the progeny and the NSC to preserve the ancestor cell. Namely, in the resting state, the Dll1-expressing granule cells surrounding the NSC would promote NSC quiescence, so only a few stimulated NSCs would undergo division rather than the whole population. Once the NSC starts to divide in response to a given stimulus, the local accumulation of the Jag1-expressing ANP progeny would gradually shift the balance of Notch signaling

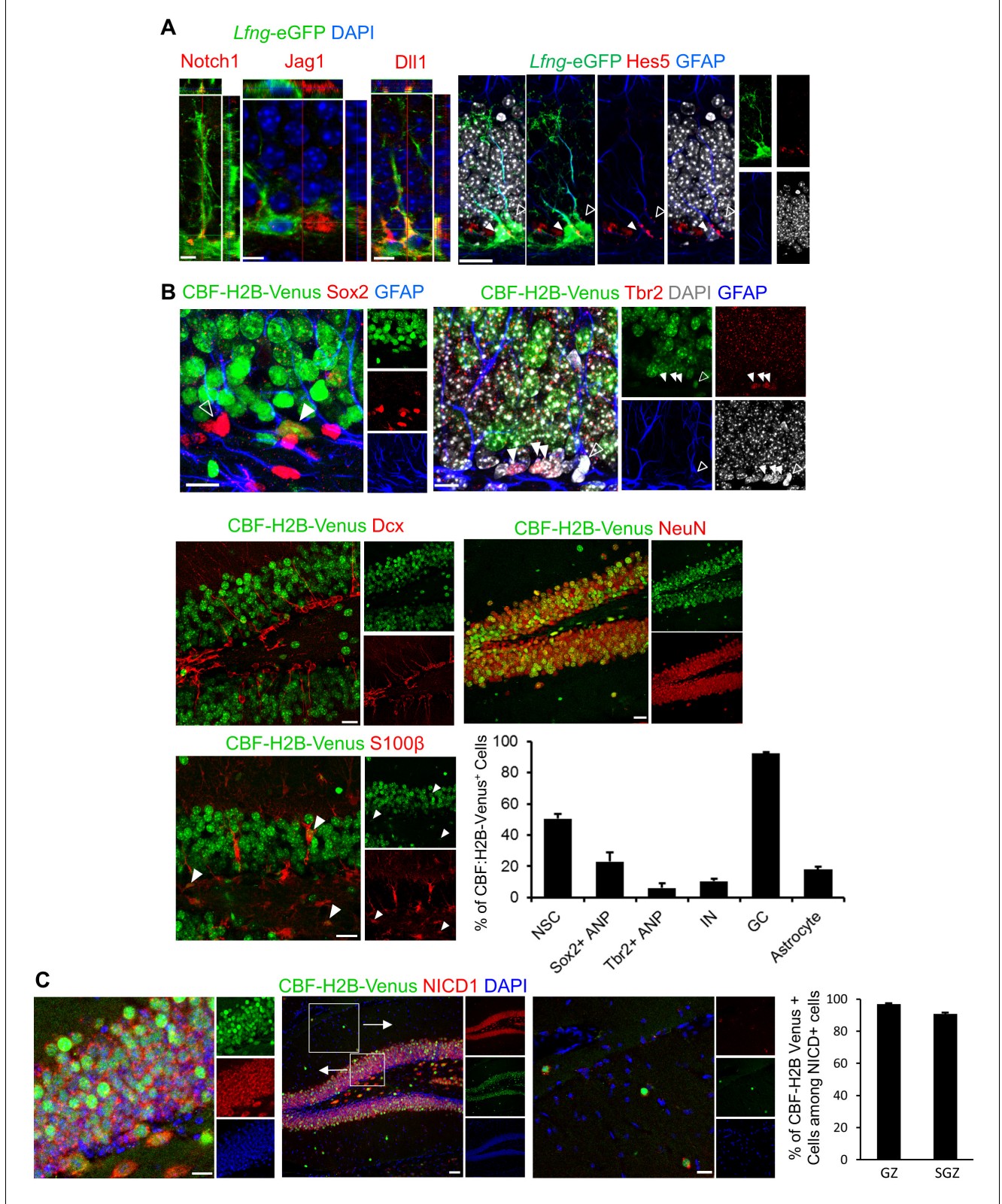

**Figure 4.** Notch pathway elements are present in the SGZ NSC niche. (**A**) Notch1 is expressed in *Lfng*-eGFP NSCs, facing granule cell layer, where late ANPs (type 2b cells) and granule cells are located. Jag1 is expressed in adjacent ANP, and Dll1 on adjacent granule cell-NSC boundary. Hes5, downstream target of canonical Notch signaling pathway, is present in some (arrowhead) but not all (empty arrowhead) NSCs. *Scale bar* = 10 μm (Notch1, Jag1, Dll1); 20 μm (Hes5). (**B**) Venus is expressed in some (arrowhead) but not all (empty arrowhead) NSCs (Sox2+ cell body and GFAP+

*Figure 4 continued on next page*

Figure 4 continued

process) in CBF:H2B-Venus mice. Type 2b cells (Tbr2) and most of the neuroblasts and immature neurons are Venus -, whereas almost all granule cells (NeuN[+]) are Venus+ with various degrees of intensity. Some of the mature astrocytes (S100$\beta^+$) have active Notch signaling (arrowheads). Quantification of Venus signal among various cell types reveals that most of the differentiating cells are devoid of active Notch signaling. As soon as the neurogenic differentiation finishes, Notch signaling is turned on again in the granule cells. *Scale bar* = 10 μm upper panels and 20 μm middle and lower panels. (C) NICD1 and Venus colocalize in CBF:H2B-Venus mice. NICD1 almost completely overlaps with the Venus signal in the SGZ (left inset), while it is lacking in the molecular layer (right inset). *Scale bar* = 20 μm. Quantification of Venus[+] cells among NICD1[+] cells in GZ and SGZ verifies high degree of colocalization (N = 3; 90.79 ± 0.77% and 96.95 ± 0.57%, respectively).

within the 'mother' NSC, eventually halting Notch signaling and leading the NSC to exit the cell cycle. This would be beneficial on two fronts: it would prevent the niche from becoming overwhelmed by the perpetual production of ANPs, and it would preserve the mother NSC from exhaustion as there is only a finite number of divisions it can endure.

## Lfng preserves NSCs by controlling their cell cycle

We first postulated that *Lfng* functions to preserve NSCs in both resting and active states, saving them from excessive mitosis. If this is the case, then lack of *Lfng* would lead to increased NSC division and their premature depletion from the niche. To examine the effect of *Lfng* ablation on NSC maintenance, we obtained *Lfng*[tm1Grid] mice (*Lfng*[−/+]; JAX 010619), in which most of the exon1 region of the *Lfng* is replaced by a targeting vector containing *β-gal* (*Zhang and Gridley, 1998*). Mice heterozygous for *Lfng* mutation are grossly normal. Homozygote mice typically have shortened trunks and malformed rib cages that impair respiration and cause premature death; the phenotype is variable, however, and some less severely affected homozygous mice do survive to adulthood (*Zhang and Gridley, 1998*). Because we could not consistently mate *Lfng*[−/+] with *Lfng*-eGFP mice, we first developed a reliable method to quantify NSCs in the absence of eGFP reporter. We used a combination of markers known to be expressed in NSCs: Sox2 for labeling the NSC cell body (*Suh et al., 2007*) and GFAP for labeling the NSC radial process (*Encinas and Enikolopov, 2008*). We compared the NSC number in wild-type mice (using GFAP/Sox2 immunostainings) and *Lfng*-eGFP mice (using eGFP expression, GFAP[+] radial process and cell morphology) and found no significant difference between the two methods (N = 3; p=0.99; *Figure 5A*). Thus, for all subsequent studies where NSCs could not be identified because of the lack of fluorescent reporter, we used the GFAP/Sox2-based identification.

Mice lacking *Lfng* had markedly fewer NSCs than wild-type mice (*Figure 5B*). This reduction in NSC number could be due to changes in the NSC cell cycle (*Ables et al., 2010*; *Breunig et al., 2007*; *Ehm et al., 2010*; *Giachino and Taylor, 2014*), as it has been suggested that NSCs undergo a finite number of divisions followed by their transformation into astrocyte-like cells (*Encinas et al., 2011*). We examined the ratio of cycling Ki67[+] NSCs and BrdU[+] NSCs in mutant and wild-type mice and found that NSCs lacking *Lfng* were both cycling (*Figure 5C*, left panel) and in S-phase (*Figure 5C*, middle panel) in significantly higher proportions than wild-type NSCs (N = 4; p=0.0221 for Ki67[+] NSCs; p=0.0017 for BrdU[+] NSCs).

Because these data were obtained in constitutive knockout mice, however, it is possible that they do not reflect the direct effect of *Lfng*. Therefore, to specifically delete *Lfng* from Lfng-expressing NSCs, we crossed *Lfng*[flox/flox] mice (*Xu et al., 2010*) with the *Lfng*-CreER[T2]; RCL-tdT mice. In the resulting conditional knockout mice (i*Lfng*[fl/fl]), the induced NSCs express non-functional mutant Lfng protein because exon 2 of the *Lfng* has been deleted (*Xu et al., 2010*, *2012*). These mutant clones can be visualized as they express tdTomato reporter. We then compared the relative number of BrdU[+] NSCs between *Lfng*-CreER[T2];RCL-tdT control mice (tdTomato[+] GFAP[+] Sox2[+]), i*Lfng*[fl/fl] wild-type clones (tdTomato[-] GFAP[+] Sox2[+]), and i*Lfng*[fl/fl] mutant clones (tdTomato[+]). Considerably more mutant NSCs were in S-phase than were either wild-type NSC clones or NSCs in control mice (*Figure 5C*, right panel). Interestingly, i*Lfng*[fl/fl] wild-type clones had a similar ratio of BrdU[+] NSCs as control mice (*Figure 5C*, right panel), suggesting that the effect of Lfng in NSCs is cell-autonomous.

These findings from both constitutive and conditional knockout models suggest that in the absence of *Lfng*, many more NSCs divide. This effect persists not only in physiological but also in pathological conditions: when *Lfng*[−/+] mice were subjected to ECS, they had a significantly higher

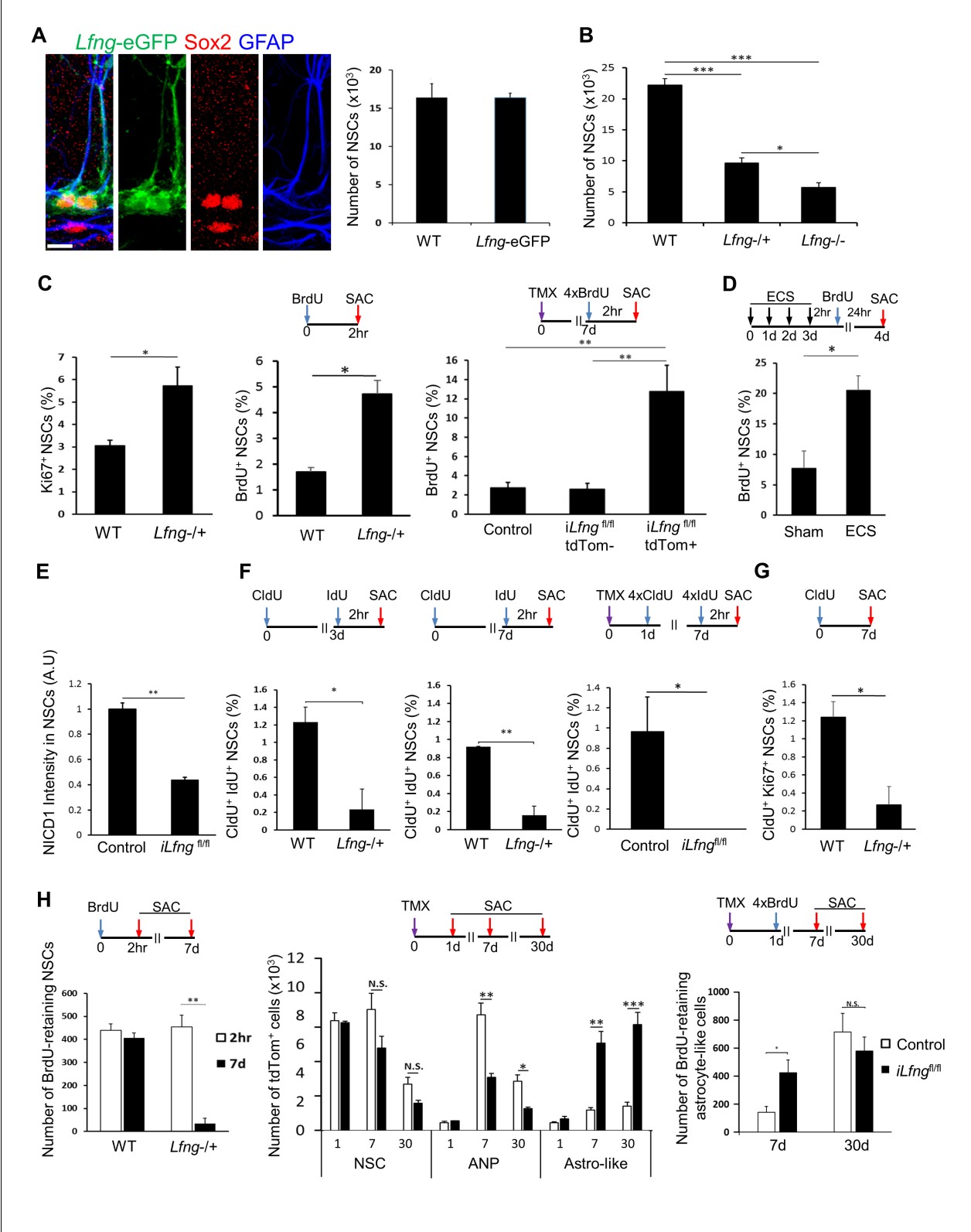

**Figure 5.** Lfng preserves NSCs by controlling their cell cycle. (**A**) *Left panel: Lfng*-eGFP NSCs express GFAP[+] radial processes originating from the Sox2[+] cell nuclei located in the SGZ. *Right panel*: The total number of GFAP[+] Sox2[+] NSCs in 4-month-old wild-type mice does not differ from the total number of eGFP[+] NSCs in 4 month-old *Lfng*-eGFP (N = 3 per group; Student's *t*-test, p=0.99). *Scale bar* = 10 μm. (**B**) The total number of NSCs in 2-month-old wild-type, *Lfng* heterozygote (*Lfng*[−/+]), and homozygote (*Lfng*[−/−]) knockout mice shows *Lfng* dose-dependent decrease in the NSC

*Figure 5 continued on next page*

Figure 5 continued

population (N = 3–4 per group; One-way ANOVA p<0.00001, Tukey HSD post-hoc test: p<0.0001 for wild-type vs $Lfng^{-/+}$ or $Lfng^{-/-}$, p=0.0484 for $Lfng^{-/+}$ vs $Lfng^{-/-}$). (C) Lack of $Lfng$ promotes increased division of NSCs. *Left panel*: NSCs lacking $Lfng$ have a higher ratio of cycling Ki67$^+$ NSCs compared to wild-type (N = 4 per group; Student's *t*-test p=0.0221). *Middle panel*: NSCs lacking $Lfng$ have a higher ratio of actively dividing, BrdU$^+$ NSCs compared to wild-type (N = 4 per group; Student's *t*-test p=0.0017). *Right panel*: $Lfng$ acts cell-autonomously and in a dose-dependent manner to control the NSC division (N = 4 per group). The ratio of BrdU$^+$ NSCs was compared between i$Lfng^{fl/fl}$ mutant clones (tdTomato$^+$), i$Lfng^{fl/fl}$ wild type clones (tdTomato$^-$ GFAP$^+$ Sox2$^+$), and $Lfng$-CreER$^{T2}$; RCL-tdT control mice (tdTomato$^+$ GFAP$^+$ Sox2$^+$). N = 4; p=0.9978 for control vs i$Lfng^{fl/fl}$ tdTom$^-$ clones; p=0.0049 for control vs i$Lfng^{fl/fl}$ tdTom$^+$ clones; p=0.0045 for i$Lfng^{fl/fl}$ tdTom$^-$ clones vs i$Lfng^{fl/fl}$ tdTom$^+$ clones. (D) NSCs lacking $Lfng$ are hyper-activated in response to ECS treatment (N = 3–4 per group; Student's *t*-test, p=0.0178). (E) Lack of $Lfng$ decreases Notch signal intensity in mutant NSCs. Relative intensities of NICD1 staining are significantly lower in i$Lfng^{fl/fl}$ mutant NSCs compared to control (N = 4 for control; N = 3 for i$Lfng^{fl/fl}$; Student's t-test, p=0.0004). (F) NSCs lacking $Lfng$ spend less time in the active state than wild-type NSCs. $Lfng$ absence is associated with decreased S-phase re-entry 3 (*left panel*) and 7 (*middle panel*) days following the initial division compared to the wild-type NSCs (N = 4 per group; Student's *t*-test, p=0.0024 for 3d, p=0.0003 for 7d). CldU$^+$ IdU$^+$ cells represent NSCs that underwent first division at the time of CldU injection (day 0) and were in S-phase at the time of IdU injection (day 3 or day 7). *Right panel*: In i$Lfng^{fl/fl}$ mice, no NSCs were found that re-entered S-phase 7 days after the initial division. CldU$^+$ IdU$^+$ cells represent NSCs that were induced at day 0, underwent first division 1 day post-induction (CldU$^+$) and were in S-phase 7 days post-induction (IdU$^+$; N = 3–4 per group; Student's *t*-test, p=0.0014). (G) NSCs lacking $Lfng$ mostly exit cell cycle within a week of first division. CldU$^+$ Ki67$^+$ NSCs represent NSCs that are actively cycling 7 days following the CldU injection (N = 4; Student's *t*-test, p=0.0105). (H) NSCs lacking $Lfng$ give rise to astrocyte-like cells. *Left panel*: The number of BrdU-retaining NSCs 7 days after the BrdU injection is significantly reduced in $Lfng^{-/+}$ mice compared to wild-type (N = 4 per timepoint; Student's *t*-test, p=0.0003). *Middle panel*: In i$Lfng^{fl/fl}$ mice, significantly more tdTomato$^+$ astrocyte-like cells accumulate 7 and 30 days following induction compared to controls, while the number of tdTomato$^+$ ANPs significantly decreases (N = 4 per group, p=0.0003 and p<0.0001 for astrocyte-like cells, p=0.0024 and p=0.0049 for ANPs). *Right panel*: The number of BrdU-retaining astrocyte-like cells 7 days after the BrdU injection is significantly higher in i$Lfng^{fl/fl}$ mice compared to controls, but the difference is lost at 30 days (N = 4 per group; Student's *t*-test, p=0.0302 for 7 days, p=0.4412 for 30 days). Bars represent mean ± SEM. *p<0.05, **p<0.001, ***p<0.0001. See *Figure 5—figure supplement 1* for further details.

The following figure supplement is available for figure 5:

**Figure supplement 1.** *Jag1* and *Lfng* affect cell survival in the SGZ.

ratio of BrdU$^+$ NSCs than observed in either sham $Lfng^{-/+}$ control (N = 3–4, p=0.0178; *Figure 5D*) or ECS-treated $Lfng$-eGFP mice (*Figure 2B*). Loss of $Lfng$ therefore causes NSCs to be more suscep-tible to both stimulus-independent and stimulus-dependent division. Indeed, Notch signaling, as measured by NICD1 immunofluorescence intensity, was much lower in individual mutant NSC clones in i$Lfng^{fl/fl}$ mice two weeks after induction compared to controls (*Figure 5E*). This suggests that decreased Notch signaling might contribute to loss of NSC quiescence in $Lfng$ mutant NSCs.

Our findings also revealed an interesting phenomenon: the difference in the fold change of Ki67$^+$/ BrdU$^+$ NSCs in mutant mice (1.21 fold) was lower than the Ki67$^+$/ BrdU$^+$ NSCs in wild-type mice (1.81 fold). This means that most cycling NSCs are in the S-phase in the mutant mice. To delve deeper into the biology of $Lfng$ effect on NSC cell cycle, we used sequential labeling with BrdU ana-logs, 5-chloro-2-deoxy-uridine (CldU) and 5-iodo-2-deoxy-uridine (IdU) (*Breunig et al., 2007*; *Encinas et al., 2011*; *Lugert et al., 2010*). To define the initial proliferative NSCs cohort, 3-month-old wild-type, $Lfng^{-/+}$, i$Lfng^{fl/fl}$, and $Lfng$-CreER$^{T2}$; RCL-tdT control mice were injected with CldU. To mark NSCs that pass through the subsequent S-phase(s), we injected IdU 3 or 7 days after CldU. By determining the fraction of CldU/IdU double-labeled NSCs, it is possible to quantify the number of NSCs that re-entered S-phase 3 or 7 days apart (*Encinas et al., 2011*). Many double-labeled NSCs were observed in the wild-type mice, only a few were detected in the $Lfng^{-/+}$ mice at either time-point, but none were found in i$Lfng^{fl/fl}$ mice (*Figure 5F*). $Lfng$ thus not only affects the NSC cell cycle but is also critically important for NSC cell cycle re-entry. Lack of double labeling could be due to intermittent division or to a genuine exit from the cell cycle. As $Lfng^{-/+}$ mice had a decreased ratio of CldU/Ki67 double labeled NSCs 7 days following CldU injection (N = 4; p=0.0105; *Figure 5G*), these data strongly indicate that $Lfng$ mutant NSCs either dilute BrdU by multiple divisions, die, or directly differentiate/ transform into another cell type over the course of seven days.

Thus, we examined the number of NSCs that retained BrdU (*Figure 5H*, left panel). In $Lfng$ mutant mice, after a two-hour pulse, only 5.2% of the initial BrdU$^+$ NSC population retained BrdU after 7 days (N = 4 per timepoint, p=0.0003). In control mice, however, 92.4% of the initial BrdU$^+$ NSC population retained BrdU after 7 days (N = 4 per timepoint, p=0.38). Further, we observed

fewer NSCs with multiple S-phase re-entry in *Lfng* mutant mice at both 3 and 7 days (*Figure 5F*), arguing against BrdU dilution. To then rule out the possibility that NSC depletion is caused by a rise in apoptosis, we turned to the conditional i*Lfng*^fl/fl mice and quantified the number of apoptotic cells. Both immunostaining for activated caspase-3 and the ApopTag assay, which detects DNA strand breaks, indicated that i*Lfng*^fl/fl actually had fewer apoptotic cells than control (*Figure 5—figure supplement 1A,B*). Most apoptotic cells in SGZ are ANPs and differentiating neuroblasts, with very few astrocytes or NSCs (*Breunig et al., 2007*; *Sierra et al., 2010*). Notch signaling in i*Lfng*^fl/fl mice is downregulated (*Figure 5E*) and thus more apoptotic cells would be expected (*Nakamura et al., 2000*). The surprising decrease in the number of apoptotic cells could thus be due to the production of fewer ANPs in i*Lfng*^fl/fl mice.

To then determine the fate of the NSCs, we quantified tdTomato^+ NSCs (Sox2^+, single radial GFAP^+ process), ANPs (Sox2^+, GFAP^-), and astrocyte-like cells (Sox2^+, multiple GFAP^+ processes emerging from the cell body). We observed a declining trend in the number of mutant NSCs (p=0.8215; 0.11; 0.06 for 1, 7, and 30 days, respectively), and a significant decline in the number of mutant ANPs (p=0.2817; 0.0024; 0.0049 for 1, 7, and 30 days, respectively) (*Figure 5H*, middle panel). This is not surprising, as *Lfng* removal caused fewer NSCs to undergo multiple rounds of cell cycle, which could impede ANP production. Interestingly, we observed a notable increase in tdTomato^+ astrocyte-like cells at 7 and 30 days post-induction (p=0.0003 and p<0.0001, respectively). This warranted further examination to ensure that these cells are the progeny of induced mutant NSCs: we quantified the BrdU-retaining astrocyte-like cells and found a greater number of these cells in *Lfng* mutant mice, but only at 7 days (N = 4 per group, p=0.0302; *Figure 5H*, right panel). At 30 days, there was no difference compared to control mice (p=0.4412), suggesting that NSCs lacking *Lfng* differentiate more rapidly into astrocyte-like cells, depleting the NSC population more quickly.

In sum, in the absence of *Lfng*, many more NSCs divide at the population level compared to controls. *Lfng*-deficient NSCs also spend less time in the active state, produce fewer ANPs over time, and tend to differentiate into astrocyte-like cells more rapidly than controls, which depletes the NSC population prematurely. These effects appear to depend on the level of *Lfng*, as homozygous conditional knockout mice had more pronounced alterations.

## Notch ligands Jag1 and Dll1 preserve NSCs by opposing effects on their cell cycle

We then set out to examine the effects of Dll1 or Jag1 loss of function on NSC recruitment, proliferation (cycling and S-phase), cell cycle duration, and generation of their immediate progeny. Our prediction was that lack of Dll1 and Jag1 would have opposing effects on these measures. Mice lacking *Dll1* (*Dll1*^tm1Gos (JAX 002957; *Dll1*^−/+)) (*Hrabě de Angelis et al., 1997*) had far fewer NSCs than wild-type mice or those lacking *Jag1* (*Jag1*^tm1Grid (JAX 010616; *Jag1*^−/+)) (*Xue et al., 1999*) or *Lfng* (*Figure 6A*, *Figure 5B*, respectively). This was accompanied by a substantial increase in the ratio of cycling, Ki67^+ cells and BrdU^+ NSCs (N = 4 per group; p=0.0007 for Ki67, p<0.0001 for BrdU experiment; *Figure 6B*), suggesting that the dramatic decrease in the NSC population in mice lacking *Dll1* might be due to increased cycling and S-phase entry, as observed in *Lfng*^−/+ mice. Interestingly, no difference in the amount of NSCs was observed between wild-type and *Jag1*^−/+ mice (N = 4 per group; p=0.1663; *Figure 6A*), but there was an upward trend in the proportion of BrdU^+ NSCs (N = 4 per group; p=0.8393; *Figure 6B*, middle panel). There was, however, a significant increase in Ki67^+ NSCs lacking *Jag1* (N = 4 per group, p<0.0001; *Figure 6B*, left panel), suggesting that these NSCs may have a prolonged cell cycle. Indeed, the proportion of Ki67^+/BrdU^+ NSCs in *Jag1*^−/+ mice was higher compared to *Dll1*^−/+ mice (3.97 vs. 1.40 fold).

To further examine this observation and delete *Jag1* specifically from the *Lfng*-expressing NSCs and their progeny, we then crossed *Jag1*^tm2Grid (*Jag1*^fl/fl; JAX 010618) mice (*Kiernan et al., 2006*), in which exon 4 of the *Jag1* is deleted (*Kiernan et al., 2006*), with the *Lfng*-CreER^T2;RCL-tdT line. The induced mutant clones lack *Jag1* and can be visualized by tdTomato. In these conditional homozygous knockout mice (i*Jag1*^fl/fl), the ratio of BrdU^+ mutant NSCs was significantly higher than in controls (*Lfng*-CreER^T2;RCL-tdT; N = 4 per group, p=0.0028; *Figure 6B*, right panel). We hypothesized that the difference between constitutive heterozygous knockout (upward trend in BrdU^+ ratio, *Figure 6B*, middle panel), and conditional homozygous knockout mice (significant increase in BrdU^+ ratio, *Figure 6B*, right panel) could be due to Jag1's ability to control cell cycle duration but not the

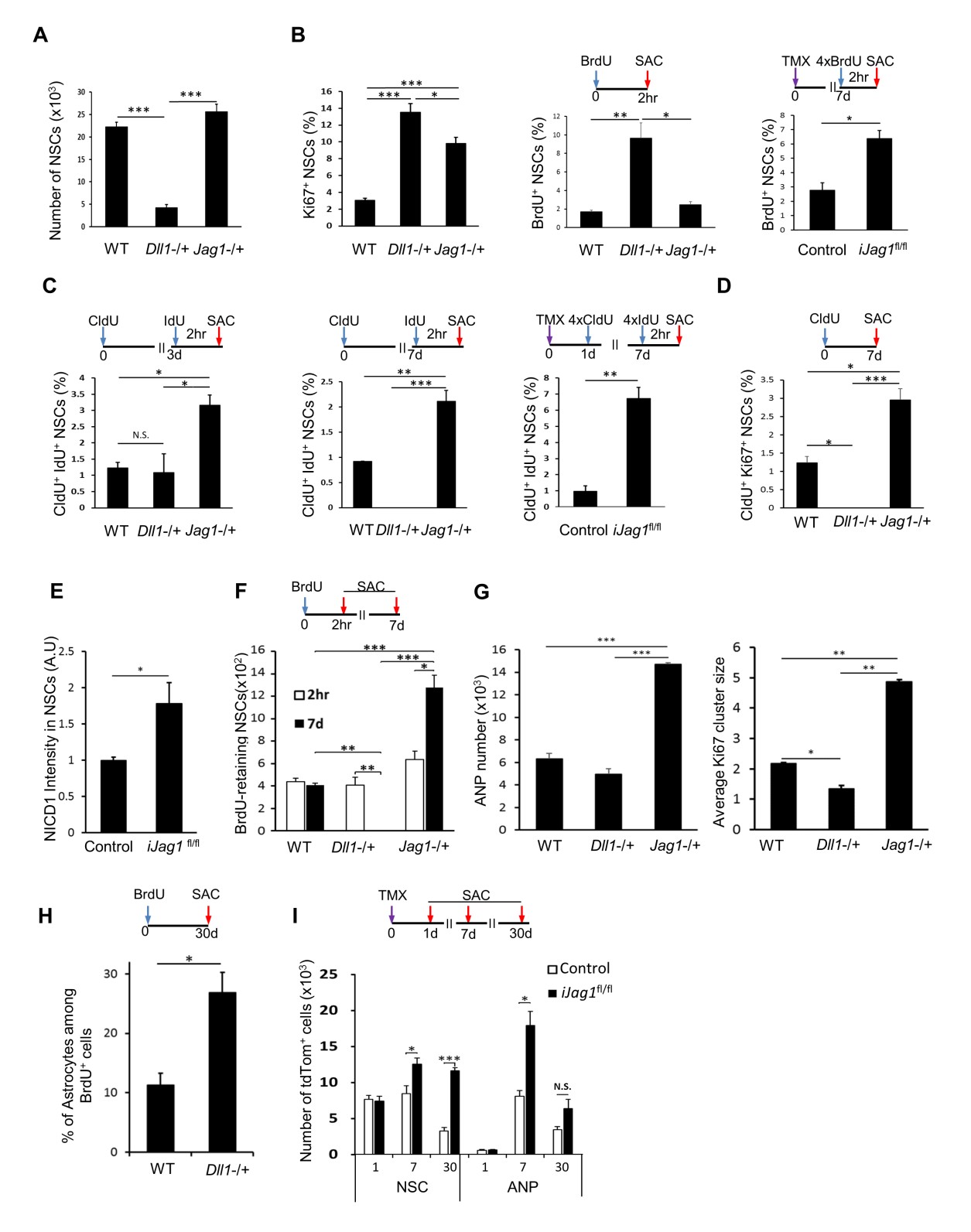

**Figure 6.** Notch ligands Jag1 and Dll1 regulate NSC cell cycle. (**A**) The number of NSCs is diminished in mice lacking *Dll1* compared to both wild-type and mice lacking *Jag1* (N = 4 per group; One-way ANOVA p<0.00001, Tukey HSD post-hoc test p<0.0001 for *Dll1* vs wild-type or *Jag1*$^{-/+}$, p=0.1663 for wild-type vs *Jag1*$^{-/+}$). (**B**) Lack of either *Dll1* or *Jag1* promotes increased division of NSCs. *Left panel*: Absence of *Dll1* or *Jag1* is associated with significantly higher proportion of cycling, Ki67$^+$ NSCs (N = 4 per group, One-way ANOVA p<0.00001, Tukey HSD post-hoc test p<0.0001 for wild-type
*Figure 6 continued on next page*

*Figure 6 continued*

vs *Dll1*$^{-/+}$ or *Jag1*$^{-/+}$, p=0.046 for *Dll1*$^{-/+}$ vs *Jag1*$^{-/+}$). *Middle panel:* Mice lacking *Dll1* have significantly higher proportion of dividing, BrdU$^+$ NSCs compared to both wild-type and *Jag1*$^{-/+}$ mice (N = 4 per group; One-way ANOVA p<0.0005, Tukey HSD post-hoc test p=0.0007 for wild-type vs *Dll1*$^{-/+}$, p=0.0015 for *Jag1*$^{-/+}$ vs *Dll1*$^{-/+}$, p=0.8393 for wild-type vs *Jag1*$^{-/+}$). *Right panel:* Absence of both copies of *Jag1* in i*Jag1*$^{fl/fl}$ mice is associated with significantly higher proportion of actively dividing, BrdU$^+$ NSCs compared to controls (N = 4 per group; Student's *t*-test, p=0.0028). (**C**) Lack of *Dll1* has an opposite effect on NSC S-phase re-entry compared to lack of *Jag1*. Relative number of NSCs that re-enter S-phase 3 (*left panel*) and 7 (*middle panel*) days following the initial division is significantly lower in mice lacking *Dll1* at 7 days, while it is significantly higher at both timepoints in mice lacking *Jag1* compared to wild-type mice (for 3d: N = 4 per group; One-way ANOVA p=0.0074, Tukey HSD post-hoc test p=0.9624 for wild-type vs *Dll1*$^{-/+}$, p=0.011 for *Jag1*$^{-/+}$ vs *Dll1*$^{-/+}$, p=0.0164 for wild-type vs *Jag1*$^{-/+}$; for 7d: N = 4 per group; One-way ANOVA p<0.0001, Tukey HSD post-hoc test p=0.0016 for wild-type vs *Dll1*$^{-/+}$, p<0.0001 for *Jag1*$^{-/+}$ vs *Dll1*$^{-/+}$, p=0.0002 for wild-type vs *Jag1*$^{-/+}$). CldU$^+$ IdU$^+$ cells represent NSCs that underwent first division at the time of CldU injection (day 0) and were in S-phase at the time of IdU injection (day 3 or day 7). *Right panel:* In i*Jag1*$^{fl/fl}$ mice, significantly higher proportion of NSCs re-enter S-phase 6 days following the initial division compared to controls (N = 4 per group; Student's *t*-test, p=0.0003). CldU$^+$ IdU$^+$ cells represent NSCs that were induced at day 0, underwent first division at 1 day post-induction (CldU$^+$) and were in S-phase (IdU$^+$) at 7 days post-induction. (**D**) Lack of *Dll1* has an opposite effect on NSC cycling time compared to lack of *Jag1* (N = 4 per group; One-way ANOVA p<0.00001, Tukey HSD post-hoc test p=0.0049 for wild-type vs *Dll1*$^{-/+}$, p=0.005 for wild-type vs *Jag1*$^{-/+}$ and p<0.0001 for *Dll1*$^{-/+}$ vs *Jag1*$^{-/+}$). CldU$^+$ Ki67$^+$ NSCs represent NSCs that are cycling 7 days following the CldU injection. (**E**) Lack of *Jag1* increases Notch signal intensity in NSCs. Relative intensities of NICD1 staining are significantly higher in *Jag1* mutant NSC clones compared to control NSCs (N = 3 for i*Jag1*$^{fl/fl}$, N = 4 for control; Student's t-test p<0.0255). (**F**) In *Dll1*$^{-/+}$ mice, no NSCs that retained BrdU 7 days after the BrdU injection were detected. In *Jag1*$^{-/+}$ mice, the absolute number of BrdU-retaining NSCs is significantly higher at both timepoints compared to wild-type mice, suggesting increased self-renewal of NSCs lacking *Jag1* (For 7d: N = 4 per group; One-way ANOVA p<0.00001, Tukey HSD post-hoc test p<0.0001 for wild-type vs *Jag1*$^{-/+}$ and *Jag1*$^{-/+}$ vs *Dll1*$^{-/+}$, and p=0.0035 for wild-type vs *Dll1*$^{-/+}$. For 2hr-7d comparisons: p=0.0028 for *Jag1*$^{-/+}$ and p=0.001 for *Dll1*$^{-/+}$ mice). (**G**) *Left graph:* Mice lacking *Jag1* have significantly more Sox2$^+$ ANPs compared to both wild-type and mice lacking *Dll1* (N = 4; One-way ANOVA p<0.00001; p<0.00001 for wild-type vs *Jag1*$^{-/+}$ and *Dll1*$^{-/+}$ vs *Jag1*$^{-/+}$; p=0.2828 for wild-type vs *Dll1*$^{-/+}$). *Right graph:* Average size of Ki67$^+$ clusters around Ki67$^+$ NSCs is larger in *Jag1*$^{-/+}$ mice compared to wild-type and *Dll1*$^{-/+}$ mice (N = 4; One-way ANOVA p<0.00001; p=0.0001 for wild-type vs *Jag1*$^{-/+}$; p=0.0015 for wild-type vs *Dll1*$^{-/+}$, p<0.0001 for *Dll1*$^{-/+}$ vs *Jag1*$^{-/+}$). (**H**) One month following BrdU injection, *Dll1*$^{-/+}$ mice have significantly higher ratio of S100$\beta^+$ progeny among newly generated cells compared to wild-type (N = 4; Student's *t*-test, p=0.016). (**I**) In i*Jag1*$^{fl/fl}$ mice, significantly more tdTomato$^+$ NSCs accumulate at 7 and 30 days post-induction compared to controls, suggesting increased self-renewal in *Jag1* mutant NSCs. This is accompanied by increased number of tdTomato$^+$ ANPs (N = 4 per group; p=0.0263 and p<0.0001 for NSCs at 7d and 30d, respectively; p=0.0033 and p=0.069 for ANPs at 7d and 30d, respectively). Please note that all wild-type and control mice presented here are same as in *Figure 5*, as the experiments using the knockout lines (*Lfng*$^{-/+}$, *Dll1*$^{-/+}$, *Jag1*$^{-/+}$, i*Lfng*$^{fl/fl}$, i*Jag1*$^{fl/fl}$) were done at the same time. The results are presented in two figures for clarity. Bars represent mean ± SEM* p<0.05, **p<0.001, ***p<0.0001. See *Figure 6—figure supplement 1* for further details.

The following figure supplement is available for figure 6:

**Figure supplement 1.** Lack of both *Lfng*$^{-/+}$ and *Jag1*$^{-/+}$ has additive effect on NSCs.

recruitment of NSCs from the quiescent population. We tested this possibility by examining the number of NSCs that were still cycling and/or in S-phase 7 days following their initial division in both *Jag1*$^{-/+}$ and i*Jag1*$^{fl/fl}$ mice.

The number of CldU/IdU double-labeled NSCs in both *Jag1*$^{-/+}$ and i*Jag1*$^{fl/fl}$ mice was notably higher than controls at both 3 and 7 days (N = 4 per group, p=0.0164 at 3d, p=0.0016 at 7d; p=0.0003 for i*Jag1*$^{fl/fl}$ mice; *Figure 6C*) and accompanied by a greater proportion of CldU$^+$ Ki67$^+$ NSCs (N = 4 per group; p=0.005 for wild-type vs *Jag1*$^{-/+}$; *Figure 6D*). Assuming a typical NSC cell cycle duration of 24–36 hr (*Brandt et al., 2012, 2010*; *Encinas et al., 2011*), these data indicate that absence of *Jag1* causes NSCs to re-enter cell cycle. The increase in NICD1 expression in individual mutant NSC clones in i*Jag1*$^{fl/fl}$ mice two weeks after induction (*Figure 6E*) supports this notion, in agreement with the reported conditional overexpression of NICD1, which increases cell cycle re-entry of NSCs (*Breunig et al., 2007*).

Conversely, there was no difference in double-labeled NSCs between *Dll1*$^{-/+}$ and wild-type mice at 3 days, but at 7 days no CldU/IdU double-positive cells could be detected, suggesting that NSCs did not re-enter cell cycle (N = 4 per group, p=0.9624 at 3d, p=0.0016 at 7d; *Figure 6C*). These observations were then verified by the CldU$^+$ Ki67$^+$ NSC analysis in *Dll1*$^{-/+}$ mice (N = 4 per group, p=0.0049; *Figure 6D*). Thus, whereas the absence of *Jag1* causes NSCs to re-enter the cell cycle while maintaining their overall number, absence of *Dll1* reduces NSC re-entry and substantially decreases their population.

To then examine the progeny of NSCs, we compared the number of NSCs that retained BrdU (BrdU$^+$ Sox2$^+$ GFAP$^+$ NSCs) in *Dll1*$^{-/+}$ and *Jag1*$^{-/+}$ vs. wild-type mice. In the absence of *Dll1*, none

of the BrdU$^+$ NSCs two hours following a single BrdU injection retained BrdU at 7 days (N = 4; *Figure 6F*). The number of Type 2a ANPs did not differ from controls, even though there was a declining trend accompanied by significantly decreased ANP Ki67$^+$ cluster size around the NSCs (*Figure 6G*). To determine the fate of NSCs in *Dll1*$^{-/+}$ mice, we injected BrdU and sacrificed the animals 30 days later. We observed significantly more S100$\beta^+$ astrocytes among the BrdU$^+$ cells. Thus, as in *Lfng*$^{-/+}$, more *Dll1* mutant NSCs terminally differentiate into astrocytes (N = 4 per group, p=0.016; *Figure 6H*).

The findings were opposite in mice lacking *Jag1*: the number of BrdU retaining NSCs was much greater 7 days following the BrdU pulse (1.6 fold, p=0.0028) than two hours after the pulse (*Figure 6F*). Thus, NSCs lacking *Jag1* may have undergone self-renewal. We then quantified the total number of tdTomato$^+$ NSCs and ANPs in iJag1$^{fl/fl}$ at 1, 7 and 30 days following induction. Initially, both control and mutant mice had a similar number of tdTomato$^+$ NSCs and ANPs, but this number markedly increased at both 7 and 30 days in mutant compared to control mice (N = 4 per group, p=0.0263 and p<0.0001 for NSC at 7d and 30d, p=0.0033 and p=0.069 for ANP at 7d and 30d; *Figure 6I*). Interestingly, there was no difference in the number of tdTomato$^+$ NSCs between 7 and 30 days (N = 3–4, p=0.3903), whereas the number of tdTomato$^+$ ANPs dropped significantly. Removal of *Jag1* therefore might lead to an initial increase in the accumulation of NSCs, previously interpreted as an indicator of self-renewal (*Dranovsky et al., 2011*). In addition, increased S-phase re-entry, observed in NSCs from both *Jag1*$^{-/+}$ and iJag1$^{fl/fl}$ (*Figure 6C*), might also lead to increased production of ANPs. Indeed, both the number of Type 2a ANPs and the size of the ANP clusters were significantly greater in *Jag1*$^{-/+}$ mice than controls (*Figure 6G*). To determine whether apoptosis had any role in the increased number of ANPs in *Jag1* mutants, we quantified the number of apoptotic cells using activated caspase-3 staining and ApopTag. Both methods demonstrated a prominent decrease in the number of apoptotic cells in iJag1$^{fl/fl}$ mice compared to wild-type (*Figure 5—figure supplement 1*).

Finally, we attempted to generate a double knockout of *Dll1* and *Jag1,* which would provide us with clues about the combinatorial effect of the lack of both ligands in the SGZ NSC niche. Mating the two constitutive lines did not succeed, most likely because of the lack of compensatory mechanisms in these two constitutive mutations during embryogenesis. As our data indicate that lack of *Lfng* or *Dll1* exhibit similar phenotypes with respect to NSC recruitment and cell cycle duration (*Figure 5B,C,F,G* and *Figure 6A–D,F*), we generated a *Lfng*$^{-/+}$; *Jag1*$^{-/+}$ double heterozygous constitutive knockout line and measured NSC number, short term (2 hr) BrdU incorporation and 1 week cell cycle re-entry (CldU$^+$/IdU$^+$) in 6 month old mice. *Lfng*$^{-/+}$; *Jag1*$^{-/+}$ double knockout mice had far fewer NSCs than wild-type (N = 3 for Lfng$^{-/+}$; Jag1$^{-/+}$ and N = 4 controls; p<0.0001 for wild-type vs Lfng$^{-/+}$ and Lfng$^{-/+}$; Jag1$^{-/+}$), but did not differ significantly from *Lfng*$^{-/+}$ mice (N = 3 for Lfng$^{-/+}$, p=0.6869 for Lfng$^{-/+}$ vs Lfng$^{-/+}$; Jag1$^{-/+}$; *Figure 6—figure supplement 1A*). The double knockout showed a higher BrdU$^+$ NSC ratio than wild-type (N = 3; p=0.0001 for wild-type vs Lfng$^{-/+}$, p=0.0003 for wild-type vs Lfng$^{-/+}$; Jag1$^{-/+}$, p=0.6724 for Lfng$^{-/+}$ vs Lfng$^{-/+}$; Jag1$^{-/+}$; *Figure 6—figure supplement 1B*), similar to *Lfng*$^{-/+}$ mice. This suggests that in both mouse models, the NSC population is exhausted by increased recruitment due to the lack of *Lfng*. Finally, we observed greater NSC cell cycle re-entry in double knockout mice (N = 3; p<0.00001 for wild-type vs Lfng-/+; Jag1$^{-/+}$ and Lfng$^{-/+}$ vs Lfng$^{-/+}$; Jag1$^{-/+}$), suggesting that removal of *Jag1* can reverse the low cell cycle re-entry phenotype observed in the NSCs lacking *Lfng* (*Figure 6—figure supplement 1C*). Overall, these data verified that *Lfng* functions primarily in the recruitment of quiescent NSCs, whereas *Jag1* functions in the cycling NSCs.

Altogether, these data indicate that Notch ligands, Dll1 and Jag1, are important regulators of NSC recruitment, cell cycle duration and exit, and generation of NSC progeny. Lack of *Dll1* increases NSC recruitment, leading to overall larger proportion of dividing NSCs compared to controls, and causes dividing NSCs to be active for a shorter period of time —there is a tendency to produce fewer ANPs and the total number of NSCs falls dramatically following increased terminal astrocytic differentiation. This phenotype thus resembles premature exhaustion of NSCs. On the other hand, lack of *Jag1* does not affect NSC recruitment, as the proportion of dividing NSCs did not differ from controls. Those that were dividing were active for a longer period and had increased cell cycle re-entry, ultimately leading to greater ANP production and increased NSC self-renewal.

## Discussion

In this paper, we demonstrate that *Lfng*, a known regulator of Notch signaling, is selectively expressed in adult NSCs, where it participates in their preservation by regulating their cycling. In addition, we provide evidence that distinct modes of Notch signaling from NSC derivatives expressing Jag1 and Dll1 ligands, most likely mediated by Lfng, directly affect the NSC cell cycle. The differential effects of NSC progeny on the ancestor cell could be the key for preserving NSCs and thus neurogenesis throughout life.

These data lead us to propose a model in which Notch signaling provides direct communication between the NSC and its descendants, such that the progeny send feedback signals to the mother cell to modify its cell cycle status (*Figure 7*). *Lfng* is central to this model, as it enables NSCs to distinguish between Dll1-expressing granule cells and Jag1-expressing ANPs. In the Notch signal-receiving NSC, Lfng glycosylation of the Notch receptor potentiates Notch signaling from Dll1 and attenuates signaling from Jag1: even though Lfng glycosylation of Notch does not affect Jag1 binding, the NICD is not cleaved from the Jag1-Notch complex. Consequently, Notch signaling decreases. In our model, the resting NSC is surrounded by mostly Dll-expressing granule cells. Thus, most Notch receptors are saturated with Dll1, which induces proteolytic cleavage and boosts the release of NICD in the NSC. The NICD is transported to the nucleus, where it activates a set of genes that ultimately maintain the NSC in a quiescent state with the potential to enter cell cycle if needed. Lfng-mediated Dll1-Notch signaling thus keeps majority of the NSCs in the quiescent state, protecting them from random activation and division.

Once a NSC is stimulated to divide (active state), it starts to produce ANPs. The first progeny (Type 2a) do not express Notch ligands and the NSC continues to actively divide. As ANPs mature into late ANPs (Type 2b), they start to express Jag1. Jag1 binding to the Lfng-modified Notch on NSC does not generate NICD, so the local accumulation of Jag1-expressing ANP progeny around the mother NSC gradually lowers the levels of NICD within the NSC and decreases Notch signaling, eventually causing NSC to stop dividing because it is surrounded by progeny. This signaling thus preserves both the spatially-enclosed SGZ niche from an overwhelming number of newborn cells and the NSC itself from over-dividing. Note that our model describes preservation of the NSCs in two contexts: in the resting state, Lfng-boosted Dll1-mediated Notch signaling preserves the NSCs at the population level (i.e., prevents global NSC activation to undergo division), while in the context of active division, Jag1-mediated inhibition of Notch signaling preserves the NSC on the single-cell level (prevents its perpetual division). Therefore, Lfng-mediated Notch signaling could be the key for preserving NSCs in both resting and active states.

Our model is supported by our data on the outcomes of *Lfng, Dll1* and *Jag1* loss-of-function with respect to the NSC number, cell cycle properties, NICD1 intensity, and fate. Lack of *Lfng* reduced NICD1 expression in NSCs and induced many more NSCs to divide at the population level; they spend less time in the active state, produce fewer ANPs and tend to differentiate into astrocyte-like cells, eventually depleting the NSC population. Similar observations were made when Notch signaling in NSCs was inhibited by targeting different components of the pathway. Removal of either Notch1 (*Ables et al., 2010*) or Rbpj-κ (*Ehm et al., 2010*) caused the depletion of NSCs, which was attributed either to a failure of NSC self-renewal (*Ables et al., 2010*) or to an initial burst in the number of proliferating NSCs followed by their depletion (*Ehm et al., 2010*). The deletion of Rbpj-κ caused a more robust and faster decrease in the NSC population compared to the removal of Notch1, possibly due to different mechanisms of Rbpj-k and Notch1 signaling. Namely, Rbpj-κ is required not only for Notch1-mediated signaling, but also Notch signaling through other receptors (*Andersson et al., 2011*). Thus, deletion of Rbpj-κ might cause a more severe depletion of NSCs as they could not remain quiescent. On the other hand, at least in the subventricular zone Notch1 seems to be required for the actively dividing NSCs but not when they are in the quiescent phase (*Basak et al., 2012*); thus, deletion of Notch1 affects only a subpopulation of NSCs.

Our data in *Lfng* mutants resemble both models, suggesting that *Lfng* mediates signaling in both quiescent NSCs (similarly to the Rbpj-k) and active NSCs (similarly to Notch1). Both modes of *Lfng*-mediated action led to multiple rounds of NSC division, which, based on the literature, could be followed by either a return to quiescence (*Bonaguidi et al., 2011*; *Lugert et al., 2010*) or a terminal differentiation into astrocyte-like cell (*Encinas et al., 2011*). Removal of *Lfng* both diminishes the intensity of Dll-Notch signaling and augments the production of NICD following Jag-Notch binding

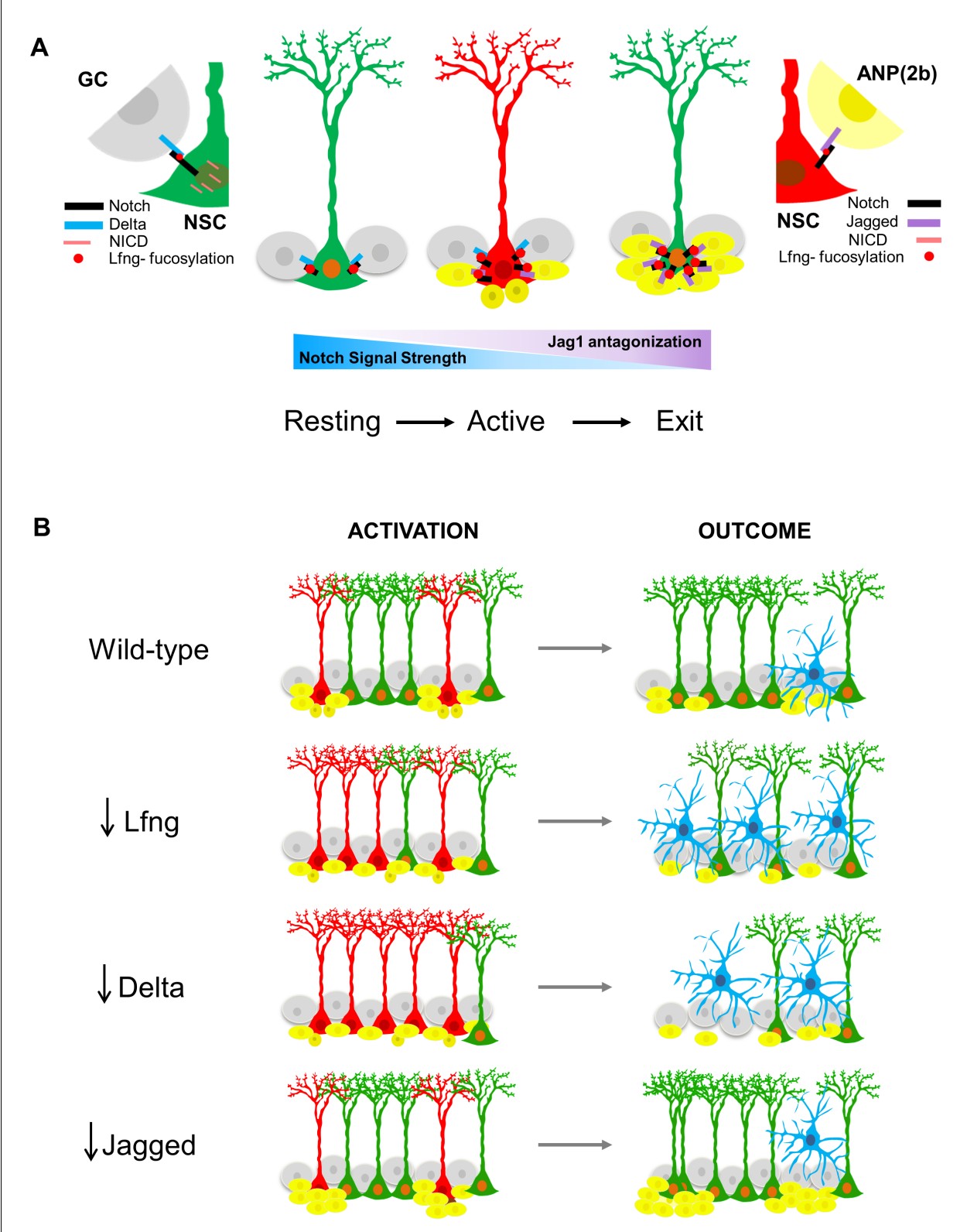

**Figure 7.** Proposed model for Lfng-mediated regulation of NSC maintenance by progeny. (**A**) In the resting state, Lfng-expressing NSC (green) is surrounded mostly by Delta1 (Dll)-expressing granule cells (GC, grey). Dll1 binds to the Lfng-modified Notch receptor on the NSC, which boosts Dll1-mediated Notch signaling by producing more NICDs and the NSC is kept quiescent but ready to undergo cell cycle if stimulated. Once activated, NSC (red) starts to produce ANPs. The first progeny (Type 2a, small dark yellow cells) do not express Notch ligands and the NSC continues to divide. As

*Figure 7 continued on next page*

*Figure 7 continued*

ANPs mature into late ANPs (Type 2b, yellow cells), they start to express Jag1. Jag1 binding to the Lfng-modified Notch receptor on the NSC does not generate NICD and thus the Notch signaling strength in the 'mother' NSC decreases. Eventually, the NSC is surrounded by mostly Jag1-expressing ANPs, and it exits active state (green NSC). Thus, in the resting state, the granule cell progeny prevents overt activation of NSCs, while in the active state, the ANP progeny prevents overt division of the NSC. These feedback signaling from the progeny both act to preserve the NSC population and the integrity of the niche. (B) The summary of the loss-of-function data, focusing on the number of activated NSCs (red) and the final outcomes of their division. Mice lacking *Lfng* and *Dll1* have similar phenotypes: NSCs are recruited in bulk, they divide less and faster, and eventually lead to depletion of NSC population. In Lfng mutant mice, there is an increased transformation into astrocytes (blue). Mice lacking *Jag1* have the opposite phenotype: while NSCs are activated as in the wild-type, they divide more and longer, and produce large clusters of ANPs as well as self-renew.

(*Benedito et al., 2009*; *LeBon et al., 2014*; *Stanley and Okajima, 2010*; *Taylor et al., 2014*; *Yang et al., 2005*). The diminished Dll-Notch signaling causes more quiescent NSCs to be recruited; once they start to produce Jag1-expressing ANPs, NSCs lacking *Lfng* will continue to be exposed to Notch signaling, which might lead to gliogenesis (*Breunig et al., 2007*; *Chambers et al., 2001*; *Imayoshi et al., 2013*; *Tanigaki et al., 2001*). Notably, we observed a greater number of astrocyte-like cells in *Lfng* mutants, not increased apoptosis. Indeed, the number of apoptotic cells in the *Lfng* mutant mice was diminished, most likely due to production of fewer ANPs.

As the Lfng-modified Notch receptor enables the cell to distinguish between Dll1 and Jag1 ligands to activate differential downstream targets (*Chapouton et al., 2010*; *Isomura and Kageyama, 2014*; *Nellemann et al., 2001*; *Ninov et al., 2012*; *Yoshiura et al., 2007*), it is not surprising that Dll1 and Jag1 have opposing effects on NSC biology. Absence of *Dll1* caused a greater proportion of NSCs to undergo division and accelerated their exit from the active state, leading to differentiation into astrocytes and premature exhaustion of the NSC population. On the other hand, absence of *Jag1* exerted its effect on the active NSC population rather than the quiescent one. We base this claim on the following: (i) the total NSC populations in $Jag1^{-/+}$ and wild-type mice did not differ, so the quiescent NSCs were not exhausted because of increased recruitment; (ii) the two-hour BrdU incorporation rate was slightly greater in NSCs lacking *Jag1* compared to controls. On the other hand, there was a significant increase in the proportion of BrdU$^+$ NSCs in i$Jag1^{fl/fl}$ mice compared to controls. This may be due to the dosage difference between the homozygous conditional and heterozygous constitutive *Jag1* knockout mice, which might result in an increase in the actively dividing NSC population rather than increased recruitment of NSCs from the quiescent pool. The extended cycling of NSCs agrees with previously reported extended cycling of NSCs that overexpress NICD (*Breunig et al., 2007*); indeed, NICD1 expression is higher in i$Jag1^{fl/fl}$ NSCs. Thus, removal of *Jag1* lowers the chance of NSC to stop dividing via sustaining the Notch signaling intensity, which suggests that Jag1 expressed on the ANP progeny is able to send feedback signals to the ancestor NSC and downregulate its Notch signaling. In addition, our data suggest that in the absence of *Jag1*, NSCs undergo self-renewal in addition to producing large amounts of ANPs. Intuitively, more ANPs should give rise to more new neurons, but conditional ablation of *Jag1* has been shown to cause aberrant neuronal lineage formation (*Lavado and Oliver, 2014*); an increase in the number of ANPs should therefore be interpreted cautiously with respect to neurogenesis. Nevertheless, increased self-renewal with no significant rise in neuronal population has been observed in social isolation (*Dranovsky et al., 2011*). Thus, it will be interesting to further investigate the mechanisms that mediate self-renewal in the future, as these processes might be exploited for targeted manipulation of NSCs.

Altogether, our data represent a substantial advance in our understanding of adult hippocampal neurogenesis, with critical implications for the preservation of adult NSCs. Namely, Lfng in NSCs along with Dll1 and Jag1 ligands in the progeny are an important part of the regulatory machinery that governs NSC maintenance, controlling their recruitment (preventing global activation), division (both the number and the termination of active state) and terminal fate (preventing over-transformation into astrocyte-like cells), both in physiological and pathological states associated with the NSC impairment.

# Materials and methods

## Animals

Experiments were performed using the following mice: *Lfng*-eGFP (RRID:MMRRC_015881-UCD) mice obtained from GENSAT (*Gong et al., 2003*). Generation of *Nestin*-GFP (MGI:5523870) and *Nestin*-CFP^nuc were described previously (*Encinas et al., 2006*; *Mignone et al., 2004*). *Lfng*-eGFP mice were received as FVB/N-C57BL/6 hybrids and crossed to C57BL/6 mice for at least 10 generations. C57BL/6J (JAX 000664; RRID:IMSR_JAX:000664), Lfng^Tm1Grid/J (*Lfng*^−/+, Lfng^β-Gal −/+, JAX 010619, RRID:IMSR_JAX:010619) (*Zhang and Gridley, 1998*), AI14 (RCL-tdT) reporter line (JAX 007908; RRID:IMSR_JAX:007908) (*Madisen et al., 2010*), CBF:H2B-Venus (JAX 020942; RRID:IMSR_JAX:020942) (*Duncan et al., 2005*), *Jag1*^tm1Grid (*Jag1*^−/+, JAX 010616; RRID:IMSR_JAX:010616) (*Xue et al., 1999*) and *Dll1*^tm1Gos (*Dll1*^−/+, JAX 002957; RRID:IMSR_JAX:002957) (*Hrabě de Angelis et al., 1997*), *Jag1*^tm2Grid (*Jag1*^flox/flox, JAX 010618; RRID:IMSR_JAX:010618) (*Kiernan et al., 2006*) were purchased from The Jackson Laboratory (Bar Harbor, ME). *Lfng*^flox/flox (*Xu et al., 2010*) mouse was a gift from Dr. Egan (Hospital for Sick Children, Toronto, ON), and is available in The Jackson Laboratory (37160-JAX). Mouse studies were approved by the Baylor College of Medicine Institutional Animal Care and Use Committee.

## Generation of the *Lfng*-CreER^T2 mouse

The RP23-270N2 BAC parent plasmid containing the *Lfng* locus was modified by recombinering to insert a CreERT2 sequence in frame at the transcriptional start site. *Lfng*-CreER^T2 mice were generated by pronuclear injection of the modified BAC construct into fertilized FVB/n embryos. To trace progeny of *Lfng*-expressing cells, the mice were crossed with Ai14 (RCL-tdT) reporter mice (*Madisen et al., 2010*).

## Immunohistochemistry

Animals were deeply anesthetized by injection of 4% Avertin per body weight and perfused transcardially with 30 ml of phosphate-buffered saline (PBS) followed by 30 ml of 4% (w/v) ice cold paraformaldehyde (PFA) in PBS. Brains were removed longitudinally; post fixed with 4% PFA for 4 hr at room temperature, and the PFA was then replaced with PBS and the tissue kept at 4°C. Free floating serial sagittal sections of 50 μm thickness were cut using a vibratome. Sections were incubated with blocking and permeabilization solution (0.3% Triton-100X and 3% BSA in PBS) for 2 hr at room temperature, followed by overnight incubation at 4°C with the primary antibody diluted in the same permeabilization-blocking solution. Sections were then washed three times with PBS and incubated with fluorochrome-conjugated secondary antibodies diluted in permeabilization-blocking solution for 2 hr at room temperature. Sections were then washed three times with PBS and mounted on coated slides with DakoCytomation Fluorescent Mounting Medium (DakoCyomation, Carpinteria, CA) as anti-fading agent.

For immunostaining against BrdU, CldU, IdU, and Ki67 sections were treated with 2N HCl for 30 min at 37°C, followed by rinsing with PBS and incubation with 0.1M sodium tetraborate (pH 8.5) for 10 min at room temperature and then again rinsing with PBS. For rabbit anti-Dll1 (Santa Cruz Biotechnology Cat# sc-9102 RRID:AB_668782) at 1:100; rabbit anti-Jag1 (Santa Cruz Biotechnology Cat# sc-8303 RRID:AB_649685) at 1:100; rabbit anti-Hes5 (Millipore Cat# AB5708 RRID:AB_11213867) at 1:100; rabbit anti-Notch1 (Santa Cruz Biotechnology Cat# sc-9170 RRID:AB_650334) at 1:100; rabbit anti-cleaved Notch1 (Assay Biotech Cat# L0119 RRID:AB_10687460 at 1:100; sections were pre-treated in 3% hydrogen peroxide solution (Sigma-Aldrich, St. Louis, MO) first at 37°C for 10 min and another 20 min at room temperature. Then sections were boiled in 10 mM citric acid-Tween (0.05%) antigen retrieval solution (pH 6) for 10 min, followed by 0.1M sodium tetraborade (pH 8.5) treatment for 10 min at room temperature. Finally, sections were incubated in permeabilization and blocking solution (1% BSA, 4% horse serum, 0.2% Triton-X) for an hour followed by overnight incubation of primary antibodies at 4°C. Anti-rabbit HRP conjugated pre-absorbed secondary antibodies were used to label these antigens for 2 hr. Finally, Tyramide Signal Amplification (TSA) Cy3 or Cy5 kit (Perkin Elmer, Waltham, MA) kit was used to amplify the signal.

In the experiments where sections were treated with HCl for BrdU, CldU, IdU, and Ki67 detection or with citric acid-based antigen retrieval solution for Dll1, Jag1, Hes5, Notch1, and NICD1

detection, we had to use antibodies against GFP or RFP to detect GFP or tdTomato, as they rapidly fade due to the acidic treatment of the tissue. The eGFP signal from *Lfng*-eGFP and *Nestin*-GFP mice, or tdTomato signal in Ai14 crossed control and conditional knockout mice was amplified with antibodies against GFP (chicken anti-GFP (Aves Labs Cat# GFP-1020 RRID:AB_10000240) at 1:1000), or with antibodies against RFP rabbit anti-RFP (Rockland Cat# 600-401-379 RRID:AB_2209751) at 1:500); goat anti-RFP (SICGEN, Cantanhede, PORTUGAL) at 1:200). Anti-GFP or anti-RFP antibodies were not used for other analyses.

For other antigens, the following antibodies were used: chicken anti-beta galactosidase (Abcam Cat# ab9361 RRID:AB_307210) at 1:1000; mouse anti-BrdU (Bio-Rad / AbD Serotec Cat# OBT0030CX RRID:AB_609566) (also used for detecting CldU) at 1:300; rat anti-BrdU (Becton Dickinson and Company Cat# 347580 RRID:AB_10015219) (also used to detect IdU); rat anti CD31 (BD Biosciences Cat# 550274 RRID:AB_393571) at 1:500; goat anti-Dcx (Santa Cruz Biotechnology Cat# sc-8066 RRID:AB_2088494) at 1:200; rabbit anti-Dcx (Cell Signaling Technology Cat# 4604S RRID: AB_10693771) at 1:200; mouse anti-GFAP (Sigma-Aldrich Cat# G3893 RRID:AB_477010) at 1:1000; rabbit anti-GFAP (Dako Cat# Z0334 RRID:AB_10013382) at 1:1000 rabbit anti-Ki67 (Vector Laboratories Cat# VP-RM04 RRID:AB_2336545) at 1:300; mouse anti-Nestin (Abcam Cat# ab6142 RRID:AB_305313) at 1:500; mouse anti-NeuN (Millipore Cat# MAB377 RRID:AB_2298772) at 1:500; mouse anti-PSA-NCAM (Millipore Cat# MAB5324 RRID:AB_11210572) at 1:400; rabbit anti-S100$\beta$ (Dako Cat# Z0311 RRID:AB_10013383) at 1:300; rabbit anti-Sox2 (Abcam Cat# ab97959 RRID:AB_2341193) at 1:500; Rabbit anti-Tbr2 (Abcam Cat# ab23345 RRID:AB_778267) at 1:000; mouse anti-Vimentin (Dako Cat# M7020 RRID:AB_2304493) at 1:300; secondary antibodies (conjugated with Alexa 488, 594, or 657) (all pre-absorbed against other species to prevent cross reactivity) (Jacskon Immunoresearch, West Grove, PA) at 1:500. Sections were counterstained with DAPI (5 µg/mL, Sigma) at 1:1000.

## Confocal microscopy and stereology

To estimate the total number of different cell types or cells that are positive for different antigens without falling into a spatial bias, 50 µm free-floating sagittal sections spanning through the whole dentate gyrus were collected in five parallel sets using vibratome. Each set contained 13–14 sections 250 µm apart from each other. Previously described modified optical dissector method (*Encinas and Enikolopov, 2008*) was used for unbiased quantification of absolute number of different cell types in the whole brain. Briefly, an observer blind to experimental groups counted immunoreactive cells that had the aforementioned markers and morphological properties in every fifth sagittal section throughout the dentate gyrus. NSCs were identified as cells with triangular cell body in the SGZ, a GFAP$^+$ radial process originating from a Sox2$^+$ nuclei in SGZ and spanning through granule cell layer in mice without the fluorescence reporter (C57BL/6, *Lfng*$^{-/+}$, *Jag1*$^{-/+}$, *Dll1*$^{-/+}$, tdTomato$^-$ clones in i*Lfng*$^{fl/fl}$, i*Jag1*$^{fl/fl}$, and *Lfng*-CreER$^{T2}$; RCL-tdT control mice). In mice with a fluorescence reporter (*Lfng*-eGFP, *Nestin*-GFP, and tdTomato$^+$ clones in i*Lfng*$^{fl/fl}$, i*Jag1*$^{fl/fl}$, and *Lfng*-CreER$^{T2}$; RCL-tdT control mice), a triangular cell body located in the SGZ and a terminal ending with fine arborizations in granule cell layer-molecular layer boundary was also included as the criteria for NSC identification. Type2a and Type2b cells were identified as GFAP$^-$ Dcx$^-$ Sox2$^+$ or Tbr2$^+$, respectively, round cells in the SGZ without a process cells. Neuroblasts and immature neurons were identified as Dcx$^+$ cells with single or multiple processes. Granule cells were identified as NeuN$^+$ cells with prominent dendritic arborizations. Astrocytes were identified as S100$\beta^+$ cells with stellar morphology. Astrocyte-like cells in i*Lfng*$^{fl/fl}$ mice were identified GFAP$^+$ S100$\beta^-$ cells with stellar morphology and multiple processes.

For quantification that required analysis of the phenotypic morphology and/or overlap of multiple markers (including BrdU, CldU, IdU, and Ki67), 20 µm thick optical sections were scanned with confocal microscope (Leica SP5, Leica SP8 or a Zeiss LSM 710). Three-dimensional reconstructions and orthogonal views were obtained using Zen 2012 SP1 software (Zeiss, Thornwood, NY) or LAS AF Lite (Leica Microsystems, Buffalo Grove, IL). Cells that were located in the uppermost focal plane were excluded from the quantification to avoid overestimation. Total counts from 13 to 14 sections were multiplied by five for the total number of cells of interest in one hemisphere and then by two to get the total number of cells for both dentate gyri. The proportion of BrdU, CldU, IdU, and Ki67 positive cells among a certain cell type was calculated by dividing the total number of positive cells to

previously calculated number of total cells of a particular genotype or to the total number of tdTomato[+] clones of a particular cell type in lineage analysis.

Clusters were defined as groups of Ki67[+] cells located around the NSC, where individual cells were separated by 20 μm from their nearest neighbor. Cells beyond this limit were counted as separate clusters. Clusters were counted in every fifth section of the dentate gyrus.

## Quantification of NICD1 signal

Following NICD1 immunofluorescence staining, multiple sections from each genotype were imaged (N = 3 for 2 weeks post TMX injection in iJag1[fl/fl] and N = 4 for control and iLfng[fl/fl] mice). 5 μm thick Z-stacks images from both CA1 region and a viewpoint that covers the whole SGZ region of the section were scanned with 2048 × 2048 pixel resolution (Zeiss710 LSM confocal microscope). Maximum intensity projections were converted to Tiff images in ZEN 2012 SP1-black edition 64 bit (Zeiss, Thornwood, NY). Regions of interest for individual NSCs (at least 20 cells per section) were drawn around in the tdTomato[+] NSC cell body located in the SGZ. Regions of interest in the whole SGZ were drawn as 15 μm thick stripes covering the whole SGZ in the section. To compensate the section-to-section differences due to staining irregularities, NICD1 immunofluorescence intensity of neurons in CA1 region (where Lfng-CreERT2 is not active) was used as control (75 μm X 50 μm rectangle). Blue channel fluorescence intensities (corresponding to Cy5 signal in TSA amplified NICD1 immunostaining) were measured by Histo tool in ZEN lite 2012-blue edition and intensities were normalized according to CA1 measurements.

## Quantification of venus/NICD1 signal overlap

Following NICD1 immunofluorescence staining, two different sections from CBF:H2b-Venus animals (N = 3) were imaged. 15 μm thick Z-stacks images with 2048x2048 pixel resolution were scanned with Zeiss710 LSM confocal microscope. At least 200 cells were counted from SGZ and GZ of each section to evaluate the overlap between Venus and NICD1 signals in the same cell.

## Detection of apoptotic cells

For activated caspase-3 staining, paraffin embedded sections (8 μm thick) from control, iLfng[fl/fl], and iJag1[fl/fl] animals two weeks post TMX injection were treated with antigen retrieval solution (Tris-EDTA pH 9.0), followed by primary antibody treatment (anti act-casp3 antibody; Abcam, Cambridge, MA) and DAB staining (Abcam, Cambridge, MA) according to manufacturer's protocol. Due to high yield clearance of apoptotic cells from SGZ (*Abiega et al., 2016*; *Li et al., 2017*; *Sierra et al., 2010*), we also used ApopTag Peroxidase in situ apoptosis detection kit (EMD Millipore, Billerica, MA). Bright field images were acquired by Zeiss Axio Imager M2 microscope and the total number of apoptotic cells located in 30 μm of SGZ was counted.

## Electroconvulsive shock (ECS) and voluntary physical exercise

ECS experiments were performed using Ugo Basile 57800 Unit (Varese-Italy). Bilateral ECS was administered via moistened pads on ear clips using pulse generator in 3-month-old male Lfng-eGFP mice (N = 4, frequency, 50 Hz, shock duration 0.5 s, pulse width 0.5msec, current 50mA) at the same time each day for four consecutive days. BrdU (150 mg/kg) was injected at the fourth day, 2 hr after the last ECS treatment. Sham animals were exposed to the same procedure but did not receive a shock. Mice were sacrificed 24 hr following the BrdU injection and brains were processed for immunostaining with anti-BrdU antibody, as described above.

For voluntary physical exercise, two animals were placed in a cage (3 cages in total, N = 6) with a running wheel and their physical activity was monitored by Actimetrics system (Wilmette, IL) for a week. Control animals were housed in the same type of cage but with a locked running wheel. BrdU (150 mg/kg) was injected at the seventh day. Mice were sacrificed 24 hr later and brains were processed for immunostaining with anti-BrdU antibody, as described above.

## Temozolomide, tamoxifen, DTX, BrdU, CldU, and IdU injections

Temozolomide (Sigma-Aldrich, St. Louis, MO, 25 mg/kg in DMSO, PBS) was administered as four intraperitoneal injections. Control mice were injected with a vehicle. Four hours after the last injection, the mice were perfused transcardially and processed as described above. Diphtheria Toxin

from Corynebacterium diphtheriae (Sigma-Aldrich, St. Louis, MO, 16 µg/kg in PBS) was administered as four intraperitoneal injections. Tamoxifen (200 mg suspended in 10 ml of 1:9 ethanol:corn oil mixture) solution was administered either at high (200 mg/kg) or low (120 mg/kg) dose. Control mice were injected with ethanol:corn oil mixture only. BrdU (150 mg/kg) was administered as described in schemes of the corresponding figures. CldU (85 mg/kg) and IdU (115 mg/kg) were administered in equimolar concentrations. BrdU and CldU were dissolved in sterile saline. IdU was dissolved in sterile saline solution that contained 2% of 0.2N NaOH.

## Statistics

Statistical analysis was performed using GraphPad Prism 7.0 (GraphPad RRID:SCR_002798). The sample size was determined based on our previous publications (*Sierra et al., 2010*, *2015*) and published data from other groups. Experiments involving two groups were compared using un-paired Student t-test. Experiments involving more than two groups with one variable were compared by One-Way Analysis of Variance (ANOVA), followed by Tukey HSD post-hoc test analysis for pairwise comparisons. Types of test were indicated in the figure legends. Exact p values were indicated in the main text next to sample size. Significance was defined as $p < 0.05$. Data are shown as mean± SEM *$p < 0.05$, **$p < 0.001$, ***$p < 0.0001$ denoted the corresponding significance levels in all graphs.

## Acknowledgements

The authors wish to thank members of the Maletic-Savatic lab for support and critical discussion of the data. We thank Onur Birol for assistance in BAC construct preparation, Dr. Roy V Sillitoe for advices on histology, Vicky Brandt for insightful editing of the paper, Drs. James Martin Paul Swinton and IBT Mouse Genetics Technology Core Laboratory for BAC injection, and Drs. Benjamin R Arenkiel, Margaret A Goodell, Christopher J Cummings, Joanna Jankowsky, and Huda Y Zoghbi for critical reviews of the first draft of the paper. The project described was supported in part by the Microscopy, RNA In Situ Hybridization and Neuropathology Core facility at Baylor College of Medicine, which is supported by a Shared Instrumentation grant from the NIH (1S10OD016167) and the NIH IDDRC grant U54HD083092 from the Eunice Kennedy Shriver National Institute Of Child Health & Human Development as well as the BCM Cytometry and Cell Sorting Core (NCRR grant S10RR024574, NIAID AI036211 and NCI P30CA125123). The content is solely the responsibility of the authors and does not necessarily represent the official views of the Eunice Kennedy Shriver National Institute Of Child Health & Human Development or the National Institutes of Health. The work was also supported in part by the Nancy Chang Award and the CPRIT grant (RP130573CPRIT) (MMS).

## Additional information

### Funding

| Funder | Grant reference number | Author |
| --- | --- | --- |
| Cancer Prevention and Research Institute of Texas | RP130573CPRIT | Mirjana Maletic-Savatic |
| Eunice Kennedy Shriver National Institute of Child Health and Human Development | U54HD083092 | Mirjana Maletic-Savatic |
| National Center for Research Resources | S10RR024574 | Mirjana Maletic-Savatic |
| National Institute of Allergy and Infectious Diseases | AI036211 | Mirjana Maletic-Savatic |
| National Cancer Center | P30CA125123 | Mirjana Maletic-Savatic |

The funders had no role in study design, data collection and interpretation, or the decision to submit the work for publication.

## Author contributions

FS, Conceptualization, Data curation, Formal analysis, Supervision, Validation, Investigation, Visualization, Methodology, Writing-original draft, Writing-review and editing; WT-SC, Data curation, Formal analysis, Investigation, Methodology; AB, Data curation, Formal analysis, Supervision, Methodology; AT, Project administration; JME, Supervision, Visualization, Methodology, Writing - review and editing; FD, NS, Resources, Methodology; AKG, Resources, Methodology, Writing-review and editing; MM-S, Conceptualization, Resources, Formal analysis, Supervision, Funding acquisition, Investigation, Methodology, Project administration, Writing-original draft, Writing-review and editing

## Author ORCIDs

Fatih Semerci, http://orcid.org/0000-0002-0512-1827
Andrew K Groves, http://orcid.org/0000-0002-0784-7998
Mirjana Maletic-Savatic, http://orcid.org/0000-0002-6548-4662

## Ethics

Animal experimentation: Mouse studies were approved by the Baylor College of Medicine Institutional Animal Care and Use Committee.

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
