## [Decision Letter]

[Editors’ note: a previous version of this study was rejected after peer review, but the authors submitted for reconsideration. The first decision letter after peer review is shown below.]

Thank you for choosing to send your work entitled "Hippocampal neural stem cells selectively express lunatic fringe" for consideration at *eLife*. Your full submission has been evaluated by a Senior Editor a Reviewing Editor and three peer reviewers, and the decision was reached after discussions between the reviewers. We regret to inform you that your work will not be considered further for publication.

There are a number of major issues raised by your reviewers and the editors handling your manuscript, but two key issues stand out: 1) the lack of lineage tracing of Lfng^+^ cells to demonstrate that these marked cells give rise to neurons in vivo and 2) the lack of substantial mechanistic insights as to how Lfng affects neurogenesis of hippocampal stem cells. As such, although the expression patterns of Lfng^+^ are provocative and this will no doubt be of interest to stem cell biologists, the critical evidence necessary to make a case for *eLife* is missing. Although the decision of *eLife* is not to publish, you may want to consider in revising your submission elsewhere that two of your reviewers found the writing difficult to read and confusing in parts.

*Reviewer #1:*

In this study the authors characterize the expression of Lunatic Fringe (Lfng) as a neural stem cells (NSCs) marker. In particular, they create a new transgenic mouse strain harboring the GFP under the control of the Lfng promoter and compare the Lfng-GFP expression with another well-established NSC reporter Nestin-GFP. They claim that, compared to Nestin-GFP, Lfng is a more specific marker for the quiescent NSC typeI. They phenotypically characterized the Lfng positive cells for other NSCs markers and demonstrate that Lfng is specifically expressed in the quiescent NSCs, being loss in the transition to amplifying neural progenitors (ANP) and derivatives. Moreover, by crossing the Lfng reporter with a Lfng knockout mouse they demonstrate that Lfng has an important role in the NSCs biology by preserving the NSCs population by restraining their proliferation. The general purpose of the study seems to be the creation and validation of a new NSCs marker useful for further studies. However, the authors present here some data about the specific role of this gene in the physiology and behavior of the NSCs.

The authors identified and counted different cell types such as NSCs and ANPs based only on cell morphology. This is an imperfect assay to discriminate the two population, the authors should include markers analysis such as Tbr2 for ANPs.

The in vitro characterization of Lfng-GFP cells should be improved. The authors isolate the GFP positive cells from Lfng-GFP mice and place them in neurosphere culture. The neurospheres lost the expression of the GFP and the authors claim that this is because the neurospheres are populated with progenitors. The authors must substantiate this claim with specific markers. Moreover, the Lfng-derived neurospheres should be analyzed for other stem cells marker expression, self-renewal and differentiation capacity. A full demonstration that these cells are NSCs is central and still needed.

In the BrdU assay described in Figure 5 is unclear how the authors determined and counted the different cell subpopulation.

The authors argue that Lfng has a role in the stem cells pool maintenance by regulating stem cells quiescence. But, at face value (see Figure.7D-E), the data show that, 1) endogenous LNFR is required to inhibits proliferation as measured by BrdU (at least on this putative NCS population above identified); 2) to promote proliferation as measured by Ki67 (in absolute terms); 3) to sustain NSC, that decline in the KOs (a data perhaps at odd with point 1); iii) to promotes differentiation. They comment that “elevated proportion of proliferating NSCs in Lfng^-/+^ mice did not lead to increased number of INs, as one may expect. Rather, the total number of INs highly correlated with the absolute number of NSCs (R=0.9948) and not the total number of proliferating NSCs.” (before Discussion).

I find this very confusing. And the following conclusion is, in my opinion, a stretched interpretation of a complex set of data: …“Together, these data indicate that loss of Lfng causes impaired NSC maintenance due to increased proliferation rates of NSCs, which over time result in decreased number of newborn cells and the phenotype indicative of premature exhaustion of neurogenesis.”…

The authors never considered other interpretations, such as cell death (by various means), transition to different cell types other than those here studied. At the very least the conclusion should be changed.

*Reviewer #2:*

In this manuscript, Semerci et al. reported the identification of Lunatic Fringe (Lfng) as an adult hippocampal neural stem cell (NSC) marker. The authors carefully characterized Lfng-eGFP cells, and compared and contrasted them with Nestin-eGFP cells. Furthermore, the authors took advantage of Lfng knockout mice to demonstrate that Lfng is required to maintain adult hippocampal NSCs. The logic of the study is clear and straightforward. The data is convincing.

I have the following comments:

1) The study focused on adult hippocampal NSCs. In Figure 1—figure supplement 1, the authors showed that Lfng-eGFP cells are present in SVZ. It is unclear whether Lfng plays a similar role in other adult NSCs, i.e. SVZ NSCs. The author should clarify this.

2) In Figure 2, the authors presented the evidence that Lfng-eGFP is expressed selectively in NSCs. It would be important to provide quantitative analysis for these experiments. Does Lfng-eGFP label all NSCs or a subpopulation? In addition, it appears that Lfng-eGFP cells include both NSCs with a triangular shape cell body and a radial process and ANPs with horizontal processes (Figure 2). Based on the examples shown in Figure 2, the authors concluded that the eGFP+ ANPs are newborn with residual eGFP. How representative are these examples? In Figure 2, it appears the vast majority of Nestin-CFP cells express some Lfng-eGFP.

3) Related to the previous comment, how does Lfng-eGFP compare with Hes5-eGFP, which preferentially labels a subset of NSCs and ANPs? The authors may want to discuss about this.

4) In the Abstract, the authors wrote "NSCs are dispensable." Does this mean for viability?

5) It is somewhat confusing who made Lfng-eGFPTg mice, authors or GENSAT? In the main text of the manuscript, it says that the authors made the mice, but in the Materials and methods it says that it was from GENSAT.

6) In Figure 5, it is hard to see any neurospheres. A better image should be provided.

*Reviewer #3:*

This paper proposes to have identified a new marker (lunatic fringe – Lfng) for neural stem cells in the sub granular zone of the dentate gyrus. While localization and morphology place Lfng in the right place at the right time and evidence is provided that Lfng cells can divide, critical evidence that their progeny give rise to neurons is lacking. In general, the paper is poorly written and there is virtually no information on the mechanism by which Lfng might affect neurogenesis of hippocampal stem cells.

The so-called bioinformatics approach does not involve any new datasets. Rather the authors simply queried published gene expression databases such as Allen and GENSAT, which provides detailed gene expression data at cellular resolution. The data in Figure 1 (supplemental?) are not publishable as they are available on the Gensat website.

There are many crucial experiments lacking in this paper, most notably a genetic lineage trace to show that Lfng positive cells give rise to neurons. What is provided is that Lfng^+^ cells can divide (BrdU) and that cells that have undergone division (BrdU^+^) can give rise to neurons. Another issue that is not addressed is the timing of expression of Lfng^+^ cells – the authors only show the timing of EGFP, which is likely very different than the timing of LFNG, given the perdurance of EGFP. The neurosphere experiments are concerning for many reasons, there is no indication of the efficiency of purification and since the Egfp transgene is not expressed, it is not valid to count the number of BrdU labeled cells and assumed they express Lfng. There is no measurement of Lfng RNA and protein in the crosses between the Gensat line and the loss of function hets. The identification of cell types is poorly controlled. Finally, there are no experiments to provide insight as to the mechanism of action, if any, of Lfng in adult hippocampal neurogenesis.

[Editors’ note: what now follows is the decision letter after the authors submitted for further consideration.]

Thank you for resubmitting your work entitled "Lunatic fringe-mediated Notch signaling regulates adult hippocampal neural stem cell maintenance" for further consideration at *eLife*. Your revised article has been favorably evaluated by a Senior Editor, a Reviewing Editor, and three reviewers.

The manuscript has been improved but there are a few remaining issues that need to be addressed before acceptance, as outlined below:

Reviewer #2 suggests a few revisions to the figures and then one simple staining. Reviewer 3 has some concerns regarding your interpretations, and these seem reasonable. We'd thus ask you to adjust the text accordingly. In addition, like reviewer 2, this reviewer asks for more stainings and demonstration that you can reliably identify Notch signal activation (reliability of Hes5 staining, NICD staining, and Notch reporter in the tissues) and also confirm the specificity of Dll1 and Jag1 antibodies (staining patterns in your knockout animals) and demonstrate which cells are high for Notch signaling and which cells are negative for Notch signaling. Similar Notch signaling analysis in your inducible Lfng knockout animals seems reasonable.

The stainings should not take long for you to do, but they will go a long way in validating your findings. The revisions will also be useful both for your and our readers.

*Reviewer #1:*

The manuscript by Maletic-Savatic et al. is remarkably improved. It now provides elegant evidence that lunatic fringe is a new marker for Hippocampal NSCs and that Lfng along with the Notch ligands Jag1 and/or Δ functions in regulating cell cycle and generation of neurons from NSCs. In its revised form, this manuscript is an important advance on our understanding of mechanisms underlying adult neurogenesis in the hippocampus. I have no substantial concerns with the revised manuscript.

*Reviewer #2:*

In this revised manuscript, Semerci et al. have made an impressive effort in extending and improving their original study. In particular, they generated a new mouse line (i.e. Lfng-CreERT2) allowing a systematic in vivo lineage tracing of LFNG-expressing cells (i.e. neural stem cells, NSCs) in the adult hippocampus. Moreover, they carried out comprehensive genetic analyses to explore potential mechanisms underlying the function of LFNG in NSC maintenance. In addition, the authors improved the characterization of cell types and NSC behavior, as well as the comparison between the Lfng-GFP and Nestin-GFP lines.

The revised manuscript is greatly improved and the authors have addressed my previous concerns.

*Reviewer #3:*

Semerci et al. investigate the function of Lunatic Fringe (Lfng) in maintaining the hippocampal NSCs. Lfng is a known component in the Notch pathway. By generating two Lfng lines: Lfng-EGFP and Lfng-CreER, the authors conducted comprehensive analysis of Lfng expression in the hippocampus. With lineage-tracing and ablation, they demonstrated that Lfng marks largely the hippocampal NSCs. To investigate the function of Lfng in NSCs, they further knockout Lfng in NSCs, and compared the results with knockout of Notch ligand Dll1 and Jag1.

Overall, the results are comprehensive and interesting, and the topic is of great interest to the broad readership of *eLife*. The analyses related to Lfng expression and lineage tracing are rigorous. This said, I have some concerns regarding the authors' interpretation of Lfng, Dll1, and Jag1 phenotypes. Below I listed some questions that should be addressed before publication.

1) The analyses regarding Jag1 and Dll1 are incomplete and the conclusions are quite forced. First of all, using straight knockout to analyze adult phenotypes often complicate the interpretation due to accumulated effects since embryonic stage. The fact that Jag1^+/-^ and inducible knockout of Jag1 have different phenotypes illustrates this point. Before more cell-type specific Jag1/Dll1 mutants are generated, these data are suggestive at best. Second, it is difficult to assess from the presented image whether Notch signaling is indeed up-regulated in the NSCs: It is unclear if NICD is on the Lfng-eGFP NSCs in the represented picture, the Hes5 staining looks like punta rather than nuclear, and CBF-H2B-Venus looks positive in the majority of the cells that are *Sox2* negative (and only in one cell that is *Sox2* positive). How specific and reliable are these antibodies as readouts of Notch signaling? The authors should do the following to strengthen this part of the data since these data are closely related to whether daughter cells indeed send feedback to stem cells via Notch signaling, one of the key conclusions of the paper:

a) Demonstrate that they can reliably identify Notch signal activation (reliability of Hes5 staining, NICD staining, and Notch reporter in the tissues they are working with) and also confirm the specificity of Dll1 and Jag1 antibodies (for example, they can examine the staining patterns in their knockout animals). Perform rigorous analysis (and quantifications) to demonstrate which cells are high for Notch signaling and which cells are negative for Notch signaling.

b) Perform similar Notch signaling analysis in their inducible Lfng knockout animals, so that they can directly compare Lfng wild type and knockout NSCs and their downstream progeny and demonstrate unequivocally that Notch signal is indeed affected in the Lfng knockout NSCs (and not other cell types).

c) Explain the discrepancies between Jag^+/-^ and inducible Jag1 knockout data.

d) Significantly revise and tone down their claims about Notch signaling and the action of Notch ligand Dll1 and Jag1 unless they can demonstrate knockout data that is more cell type specific.

2) The authors' interpretation for their BrdU, or CIdU/IdU phenotype in Lfng mutant and Dll1/Jag1 is confusing and difficult to understand. Some textual changes will be essential to make these data more accessible to the readers.

Specifically, their data suggest that in Lfng mutant, (i) label retaining NSCs decrease with time (which is not due to multiple rounds of cell cycle entry during the chase period since fewer NSCs enter S phase twice in Lfng mutant); (ii) label retaining astrocytes increase with time; (iii) more NSCs are proliferating at any given time; and (iv) NSCs numbers are reduced. If apoptosis status is not altered in Lfng mutants (see below), this data together with the lineage-tracing data suggest that upon loss of Lfng, more NSCs are in cycle at any given time, but there is a shift in the division outcomes with regard to what cells are produced. In wild type, many of these divisions generate ANPs and stem cells, while in Lfng knockout animals, more astrocytes are produced.

The authors' statements like "NSCs lacking Lfng mostly exited the cell cycle", "Lfng removal caused NSCs to exit cycling mode earlier", or "these data indicate that absence of Jag1 causes NSCs to be in cell cycle for a prolonged length of time, while absence of Dll1 causes them to exit cell cycle" are confusing, and send out the opposite message. "NSCs exit cell cycle" implies that "NSCs are withdrawn from the cell cycle and become quiescent but remain as NSCs". If this is the phenotype, they should have observed more BrdU label-retaining NSCs, not less. A more appropriate statement might be "NSCs differentiate into astrocytes". In all, the authors should modify all of these related statements to convey the results more truthfully and enhance the readability of their manuscript.

3) The authors should also conduct apoptosis analysis to determine if changes in cell numbers are due to abnormal apoptosis, which is an independent mechanism that will influence interpretation.

4) The authors should provide evidence suggesting that Lfng-CreER is not active without tamoxifen induction (i.e., no leakage issue that might confound the interpretation).

---

## [Author Response]

[Editors’ note: the author responses to the first round of peer review follow.]

*Reviewer #1:*

*[…] The authors identified and counted different cell types such as NSCs and ANPs based only on cell morphology. This is an imperfect assay to discriminate the two population, the authors should include markers analysis such as Tbr2 for ANPs.*

We agree that cell morphology is imperfect; we have now done extensive quantifications based on the expression of previously known markers of these cells. First, we have established that markers known to label NSCs, such as nestin, *Sox2*, vimentin and GFAP, were all expressed in *Lfng*(BAC)::eGFP+ cells (Figure 1). ANPs were discriminated from NSCs based not only on their round morphology but also the presence of *Sox2* and Tbr2 and lack of markers such as GFAP and Dcx. We distinguished early ANPs (Type 2a: round cells in the SGZ that express *Sox2* and lack GFAP and Dcx) from late ANPs (Type 2b: round cells in the SGZ that express Tbr2) (Figure 1). We then continued to use these markers to quantify NSCs and ANPs in most other experiments throughout the paper.

To demonstrate the specificity of Lfng expression to NSCs, we performed a series of additional experiments. Using the *Lfng*(BAC)::eGFP/*Nestin*-CFP^nuc^ double transgenic mice and immunostaining with markers listed above, we detected only 7% (7.41 ± 1.95%; N=3) ANPs that were eGFP+. Next, we showed that *Lfng*(BAC)::eGFP is expressed only in minority of Type2a ANPs (*Sox2* + GFAP- cells in the SGZ; 10.58 ± 2.38, N=3), unlike *Nestin*-GFP (94.7 ± 0.34, N=4). We then determined the clearance time of the eGFP signal from Type 2a ANPs, using BrdU pulse-and-chase labeling. eGFP was cleared between days 1 and 3 (Figure 1 left graph), which correlates with the eGFP half-life. Finally, using temozolomide to kill dividing cells, we quantified NSCs and ANPs and demonstrated that Lfng-expressing NSCs were mostly quiescent (Figure 2). Together, these data strongly suggest that most of the *Lfng*(BAC)::eGFP+ cells are NSCs. The *Lfng*(BAC)::eGFP+ ANPs are the immediate progeny of NSCs that retained some eGFP due to sequestration of the protein during cell division. ANPs quickly lose eGFP, implying that *Lfng*(BAC)::eGFP expression is selective for NSCs and that *Lfng* is active in NSCs but not in ANPs.

*The in vitro characterization of Lfng-GFP cells should be improved. The authors isolate the GFP positive cells from Lfng-GFP mice and place them in neurosphere culture. The neurospheres lost the expression of the GFP and the authors claim that this is because the neurospheres are populated with progenitors. The authors must substantiate this claim with specific markers. Moreover, the Lfng-derived neurospheres should be analyzed for other stem cells marker expression, self-renewal and differentiation capacity. A full demonstration that these cells are NSCs is central and still needed.*

We have now generated the lineage tracing mouse and effectively demonstrated the multipotentiality of Lfng expressing cells in vivo (Figure 3 and Figure 3—figure supplement 1). Therefore, we have removed the in vitro data from the revised paper.

*In the BrdU assay described in Figure 5 is unclear how the authors determined and counted the different cell subpopulation.*

Figure 5 in the initial submission is now Figure 2, and the quantification of the cells is described in detail in the Materials and methods section. For each time point in each genotype (*Nestin*-GFP and *Lfng*-eGFP), we performed three sets of immunostainings from each animal (n=4 per time point). We used: 1) mouse anti-GFAP, rabbit anti-*Sox2* and rat anti-BrdU for the first set; 2) rabbit anti-Dcx, mouse anti-NeuN, and rat anti-BrdU for the second; and 3) mouse anti-GFAP, rabbit antiDcx, and rat anti-BrdU for the third. NSCs were identified as BrdU^+^ cells with a *Sox2Sox2*^+^ nuclei and triangular soma in the SGZ, from which a single GFAP+ radial process extended orthogonally and spanned the granule cell layer, ending in fine arborizations within the molecular layer. These cells lacked all other markers. ANPs were identified as BrdU^+^ round GFAP- Dcx-cells in the SGZ. Immature neurons-neuroblasts and granule cells were Dcx+ and NeuN+, respectively. Cells outside of the SGZ and negative for Dcx and NeuN were categorized as 'other'.

To estimate the total number of different cell types or cells that are positive for different antigens without falling into a spatial bias, 50μm free-floating sagittal sections spanning through the whole dentate gyrus were collected in 5 parallel sets using a vibratome. Each set contained 13-14 sections 250μm apart from each other. The previously described modified optical dissector method (Encinas and Enikolopov, 2008) was used for unbiased quantification of absolute number of different cell types in the whole brain. Briefly, an observer blind to experimental groups counted immunoreactive cells that had the aforementioned markers and morphological properties in every fifth sagittal section throughout the dentate gyrus. Cells that were located in the uppermost focal plane were excluded from the quantification to avoid overestimation. Total counts from 13-14 sections were multiplied by 5 for the total number of cells of interest in one hemisphere and then by 2 to get the total number of cells for both dentate gyri.

*The authors argue that Lfng has a role in the stem cells pool maintenance by regulating stem cells quiescence. But, at face value (see Figure 7), the data show that, 1) endogenous LNFR is required to inhibits proliferation as measured by BrdU (at least on this putative NCS population above identified); 2) to promote proliferation as measured by Ki67 (in absolute terms); 3) to sustain NSC, that decline in the KOs (a data perhaps at odd with point 1); iii) to promotes differentiation. They comment that “elevated proportion of proliferating NSCs in Lfng^-/+^ mice did not lead to increased number of INs, as one may expect. Rather, the total number of INs highly correlated with the absolute number of NSCs (R=0.9948) and not the total number of proliferating NSCs.” (before Discussion).*

*I find this very confusing. And the following conclusion is, in my opinion, a stretched interpretation of a complex set of data: “Together, these data indicate that loss of Lfng causes impaired NSC maintenance due to increased proliferation rates of NSCs, which over time result in decreased number of newborn cells and the phenotype indicative of premature exhaustion of neurogenesis.”*

*The authors never considered other interpretations, such as cell death (by various means), transition to different cell types other than those here studied. At the very least the conclusion should be changed.*

In the initial paper, we showed that in the constitutive *Lfng* knockout mice, there is increased recruitment of NSCs and thus an increased ratio of both BrdU^+^ and Ki67+ NSCs. We concluded that lack of *Lfng* impairs NSC maintenance by initially increasing the rate of proliferation, which prematurely exhausts the NSC population and thus the ultimate number of newborn neurons. We agree with the reviewer that further proof was warranted.

In the revised paper, we provide additional data using both constitutive and conditional knockout mice, in which we specifically deleted *Lfng* from Lfng-expressing NSCs (Figure 4). In sum, those data demonstrate that in the absence of Lfng, many more NSCs divide at the population level compared to controls. Lfng-deficient NSCs spend less time in the active state, produce fewer ANPs over time, and have a tendency to differentiate into astrocyte-like cells more rapidly than controls, which depletes the NSC population prematurely. These effects appear to depend on the level of Lfng, as homozygous conditional knockout mice had more pronounced alterations.

*Reviewer #2:*

*[…] I have the following comments:*

*1) The study focused on adult hippocampal NSCs. In Figure 1—figure supplement 1, the authors showed that Lfng-eGFP cells are present in SVZ. It is unclear whether Lfng plays a similar role in other adult NSCs, i.e. SVZ NSCs. The author should clarify this.*

In both the initial paper and the revised manuscript, we characterize *Lfng*-eGFP NSCs in the SGZ only. We plan to characterize Lfng expression and function in SVZ niche in a future study.

*2) In Figure 2, the authors presented the evidence that Lfng-eGFP is expressed selectively in NSCs. It would be important to provide quantitative analysis for these experiments. Does Lfng-eGFP label all NSCs or a subpopulation? In addition, it appears that Lfng-eGFP cells include both NSCs with a triangular shape cell body and a radial process and ANPs with horizontal processes (Figure 2). Based on the examples shown in Figure 2, the authors concluded that the eGFP+ ANPs are newborn with residual eGFP. How representative are these examples? In Figure 2, it appears the vast majority of Nestin-CFP cells express some Lfng-eGFP.*

Please see the answer to Reviewer #1, first comment, for an outline of the methodology used to quantify and distinguish NSCs and ANPs. In the revised manuscript, we have performed additional experiments to address the reviewers’ questions.

A nearly complete overlap between *Lfng*-eGFP expression and Nestin (Figure 1) indicates that Lfng expressing NSCs are mostly the same population of NSCs labeled by *Nestin*-GFP. We corroborated this finding by comparing the total number of NSCs in *Nestin*-GFP and *Lfng*-eGFP mice at distinct time points throughout life (Figure 2). In addition, 84% of the *Lfng*-eGFP+ cells were also GFAP+ (Figure 1). As nearly all GFAP+ radial processes that originate from SGZ and end at the border between the granule cell and molecular layers were also *Lfng*-eGFP+, it is plausible that Lfng is expressed in the same population of NSCs that are labeled with *GFAP*GFP. Nevertheless, NSCs are very heterogeneous and there are differences in the labeling efficiencies depending on the transgenic mouse line (Giachino and Taylor, 2014; Semerci and Maletic-Savatic, 2016). Thus, we cannot rule out the possibility that *Lfng*-eGFP labels in part a different population of cells in comparison to other mouse models. We have done extensive work to demonstrate its specificity to NSCs and will pursue characterization of sub-populations of *Lfng*-eGFP+ NSCs in the future.

*3) Related to the previous comment, how does Lfng-eGFP compare with Hes5-eGFP, which preferentially labels a subset of NSCs and ANPs? The authors may want to discuss about this.*

To determine whether active Notch signaling is present in the *Lfng*-eGFP NSCs, we performed immunostaining against cleaved intracellular domain of Notch1 (NICD1) and Hes5 (Figure 5), as well as used the Notch reporter mouse, CBF-H2B-Venus (Duncan et al., 2005). These experiments all verified that only a subset of NSCs have active Notch signaling, indicating a similar pattern observed in *Hes5*-eGFP mice (Basak and Taylor, 2007; Lugert et al., 2012). Lack of NICD1 and Hes5 in many *Lfng*-eGFP+ NSCs suggests that Lfng expression is not limited to the cells with active Notch signaling but rather extends to a broader population.

*4) In the Abstract, the authors wrote "NSCs are dispensable." Does this mean for viability?*

In the revised manuscript, we eliminated this confusing statement and instead state that NSCs have different fates, ranging from terminal astrocytic differentiation to self-renewal.

*5) It is somewhat confusing who made Lfng-eGFPTg mice, authors or GENSAT? In the main text of the manuscript, it says that the authors made the mice, but in the Materials and methods it says that it was from GENSAT.*

We thank the reviewer for noticing the ambiguity. In the revised paper, we made it clear that we purchased the *Lfng*-eGFP mice from GENSAT as FVB/N- C57BL/6 hybrid. We then crossed it to C57BL/6 mice for at least 10 generations before starting the characterization of the *Lfng*-eGFP expression in the SGZ.

*6) In Figure 5, it is hard to see any neurospheres. A better image should be provided.*

In the revised manuscript, we excluded the in vitro studies in favor of our newer in vivo evidence of the *Lfng*-eGFP NSC multipotentiality, from the lineage tracing mouse (Figure 3 and Figure 3—figure supplement 1).

*Reviewer #3:*

*This paper proposes to have identified a new marker (lunatic fringe – Lfng) for neural stem cells in the sub granular zone of the dentate gyrus. While localization and morphology place Lfng in the right place at the right time and evidence is provided that Lfng cells can divide, critical evidence that their progeny give rise to neurons is lacking.*

We have now generated a lineage tracing mouse and provide in vivo evidence that Lfng expressing cells are multipotent stem cells, able to self-renew and give rise to both neurons and astrocytes (Figure 3 and Figure 3—figure supplement 1).

*In general, the paper is poorly written and there is virtually no information on the mechanism by which Lfng might affect neurogenesis of hippocampal stem cells.*

We have extensively revised the paper and conducted new experiments that provide new evidence on the role of Lfng in NSCs (Figure 4). Further, we also provide new data on Notch ligands, Jag1 and Dll1, and their effects on NSC cell cycle and fate (Figure 5). We also worked with a native English speaker to improve the clarity of the writing, and we hope the manuscript is now easier to read.

*The so-called bioinformatics approach does not involve any new datasets. Rather the authors simply queried published gene expression databases such as Allen and GENSAT, which provides detailed gene expression data at cellular resolution. The data in Figure 1 (supplemental?) are not publishable as they are available on the Gensat website.*

In the revised paper, we explain our database query in the Introduction, as this query has led us to hypothesize that Lfng is a selective marker of NSCs. We removed the expression data from other brain regions; please note that in the initial paper these data were not taken from the GENSAT database but were produced from our own experiments. The fact that they matched GENSAT expression data reassured us that back-crossing to C57/Bl6 mice did not affect the Lfng expression pattern.

*There are many crucial experiments lacking in this paper, most notably a genetic lineage trace to show that Lfng positive cells give rise to neurons. What is provided is that Lfng^+^ cells can divide (BrdU) and that cells that have undergone division (BrdU^+^) can give rise to neurons.*

We have now generated a lineage tracing mouse and provided the in vivo evidence that Lfng expressing cells are multipotent stem cells, able to self-renew and give rise to both neurons and astrocytes (Figure 3 and Figure 3—figure supplement 1).

*Another issue that is not addressed is the timing of expression of Lfng^+^ cells – the authors only show the timing of EGFP, which is likely very different than the timing of LFNG, given the perdurance of EGFP.*

The reviewer raises an important issue: how much *Lfng*-eGFP expression correlates with Lfng protein expression? First, Lfng has been shown to have an oscillatory expression pattern in other systems (Kageyama et al., 2007; Maroto et al., 2005; Okubo et al., 2012). We don't know if it oscillates in NSCs. This would be important if Lfng is expressed in only a subset of NSCs and not in the whole NSC population. In this case, *Lfng*-eGFP would partially label the NSC population and thus, the total number of labeled cell would be significantly different from the total number of NSCs in other mouse models, such as *Nestin*-GFP. However, our data showed that almost all *Lfng*-eGFP+ cells express Nestin (Figure 1) and further, that *Lfng*-eGFP and *Nestin*-GFP mice have similar numbers of NSCs throughout their lifetime (Figure 2). These data suggest that Lfng expression is stable enough to drive the expression of eGFP in all NSCs labeled by Nestin. Other eGFP reporter lines such as *Hes5*-eGFP (Basak and Taylor, 2007; Lugert et al., 2012) that are based on Hes5 promoter with an oscillatory expression pattern, label only a subset of NSCs in which Hes5 is present.

Second, eGFP might stay in cells longer than Lfng protein. To determine the eGFP clearance from the labeled cells, we have performed additional experiments that demonstrate that eGFP is completely lost from the cells between day 1 and day 3 (Figure 1 left graph). Thus, even if Lfng has an oscillatory expression pattern, it is shorter than the life of eGFP.

Finally, our knockout experiments, both constitutive and conditional, indicate that Lfng expression is important not only for a part of NSC life, but for both quiescent and active NSCs (Figure 4).

*The neurosphere experiments are concerning for many reasons, there is no indication of the efficiency of purification and since the Egfp transgene is not expressed, it is not valid to count the number of BrdU labeled cells and assumed they express Lfng.*

Please see the answer to reviewer #1, second comment.

*There is no measurement of Lfng RNA and protein in the crosses between the Gensat line and the loss of function hets.*

Both the constitutive and conditional Lfng knockout mice have been used in previous studies where Lfng protein levels and the RNA have been determined (Benedito et al., 2009; Xu et al., 2010; Xu et al., 2012; Zhang and Gridley, 1998). We have thus not performed these experiments.

*The identification of cell types is poorly controlled.*

In the revised paper, we have done extensive quantifications based not only on the morphology of cells but also on the expression of previously known markers of these cells, as described in detail in the methods section. Please also see the answer to reviewer #1, first comment above.

*Finally, there are no experiments to provide insight as to the mechanism of action, if any, of Lfng in adult hippocampal neurogenesis.*

We have extensively revised the paper and conducted new experiments that provide new evidence on the role of Lfng in NSCs (Figure 4). Further, we also provide new data on Notch ligands, Jag1 and Dll1, and their effects on NSC cell cycle and fate (Figure 5). These data suggest the Lfng is a key modulator of NSC cell cycle and fate.

[Editors' note: the author responses to the re-review follow.]

*Reviewer #3:*

*[…] Overall, the results are comprehensive and interesting, and the topic is of great interest to the broad readership of eLife. The analyses related to Lfng expression and lineage tracing are rigorous. This said, I have some concerns regarding the authors' interpretation of Lfng, Dll1, and Jag1 phenotypes. Below I listed some questions that should be addressed before publication.*

*1) The analyses regarding Jag1 and Dll1 are incomplete and the conclusions are quite forced. First of all, using straight knockout to analyze adult phenotypes often complicate the interpretation due to accumulated effects since embryonic stage. The fact that Jag1^+/-^ and inducible knockout of Jag1 have different phenotypes illustrates this point. Before more cell-type specific Jag1/Dll1 mutants are generated, these data are suggestive at best. Second, it is difficult to assess from the presented image whether Notch signaling is indeed up-regulated in the NSCs: It is unclear if NICD is on the Lfng-eGFP NSCs in the represented picture, the Hes5 staining looks like punta rather than nuclear, and CBF-H2B-Venus looks positive in the majority of the cells that are Sox2 negative (and only in one cell that is Sox2 positive). How specific and reliable are these antibodies as readouts of Notch signaling? The authors should do the following to strengthen this part of the data since these data are closely related to whether daughter cells indeed send feedback to stem cells via Notch signaling, one of the key conclusions of the paper:*

*a) Demonstrate that they can reliably identify Notch signal activation (reliability of Hes5 staining, NICD staining, and Notch reporter in the tissues they are working with) and also confirm the specificity of Dll1 and Jag1 antibodies (for example, they can examine the staining patterns in their knockout animals). Perform rigorous analysis (and quantifications) to demonstrate which cells are high for Notch signaling and which cells are negative for Notch signaling.*

We thank the reviewer for pointing out the discrepancy between CBF-H2B-Venus signal and NICD staining, which motivated us to search for a new antibody and optimize the protocol.

In our initial submission, we used CBF-H2F-Venus mice to demonstrate Notch activity in NSCs (Figure 4 – upper right panel). Both reviewer 2 and reviewer 3 urged us to complement these data with more detailed examination of Notch signaling through NICD staining, particularly in mutant mice.

To directly assess the presence and intensity of Notch signaling, we first need a reliable method to detect Notch signaling strength. Although CBF:H2B-Venus mice are a good tool to evaluate the presence or absence of Notch signaling, the accumulation of Venus fluorescence protein and high fluorescence intensity in the GZ (please see Figure 4) make it hard to perform reliable intensity quantifications in the SGZ where NSCs reside. We therefore turned to NICD1 as a more reliable measure of Notch signaling intensity.

To verify the reliability of NICD1 staining, we reasoned that NICD1 should be present in cells expressing Venus. Thus, we first stained CBF-H2B-Venus sections with four commercially available NICD1 antibodies (Abcam, cat#: 8925; Cell Signaling, cat#: 24215 and 4147S; and Assay Biotech, cat#L0119), optimized the staining protocol, and quantified the overlap between NICD1 staining and Venus. We found that rabbit anti-NICD1 from Assay BioTech, CA (cat#L0119) is the best available antibody in the market to show over 90% (90.79% ± 0.77%) signal overlap with CBF-H2B-Venus in SGZ and over 95% (96.95% ± 0.57%) in GZ (Figure 4 –left insetandright graph). Notably, NICD1 staining was mostly absent in the molecular layer and present in almost all of cells in which Venus was also present (Figure 4 – right inset). We then quantified the fluorescence intensity of NICD1 staining in both *iLfng*^fl/fl^ and *iJag1*^fl/fl^ mutant NSC clones (Figure 5; Figure 6). To normalize the intensity between different sections and samples, we chose NICD1 staining intensity in CA1 neurons of the hippocampus as the control, as there was no *Lfng*-driven CreERT2 activity in these cells. In *Lfng* mutant NSC clones, we expected to see a decrease in Notch signaling as Lfng potentiates Dll1-mediated Notch signals. Indeed, in *iLfng*^fl/fl^ mice, we observed a significant drop in Notch signaling intensity, suggesting that decrease in Notch signaling might contribute to loss of NSC quiescence in *Lfng* mutant NSCs (Figure 5). In *Jag1* mutant NSC clones, we expected to see an increase in Notch signaling, due to the removal of inhibitory effect of Jag1 on Lfng- modified Notch receptor. Sure enough, we detected a significant increase in Notch signaling intensity in NSCs of *Jag1* mutant mice, which might contribute to extended cycling of these cells (Figure 6). Since our *Dll1* knockout mouse model is constitutive and has uniform Dll1 expression, we do not have a control region to normalize between samples. Thus, we could not examine the NICD1 signal intensity in *Dll1*^-/+^ mice.

We present these data in Results (subsections “Notch pathway elements are present in the SGZ NSC niche”, “Lfng preserves NSCs by controlling their cell cycle” and “Notch ligands Jag1 and Dll1 preserve NSCs by opposing effects on their cell cycle”) and describe the methodology in Materials and Methods (subsection “Confocal microscopy and stereology”).

We performed a panel of stainings in the CBF-H2B-Venus mice to examine which cells have Notch signaling in the SGZ and GZ of dentate gyrus (Figure 4). As reviewer #3 points out, Venus signal as well as NICD1 staining was mostly localized to GZ, where *Sox2*^-^ cells reside (Figure 4). Notch signaling plays an important role in the maintenance of granule neurons (Zhuang et al., 2015); thus, granule cells are expected to be positive for both NICD (Brandt et al., 2010; Breunig et al., 2007) and CBF-H2B-Venus signal. Indeed, CBF-H2B-Venus signal overlapped almost completely with NeuN staining with various degrees of intensity (Figure 4). However, most Dcx^+^ cells were devoid of Venus signal (Figure 4) and only quarter of S100β^+^ astrocytes were positive for Venus (Figure 4). We agree with reviewer #3 that some of the *Sox2Sox2*^+^ cells are negative for CBF-H2B-Venus signal, suggesting that some of the progenitor cells are devoid of active Notch signaling. However, as shown in Figure 4 – right panel, B(arrow heads), some of the NSCs are positive for Notch signal. This is in agreement with findings from Breunig and colleagues, who showed that only some of the NSCs have nuclear NICD staining (Breunig et al., 2007). Quantification of Venus^+^ cells among different cell types revealed that cells with active Notch signaling progressively decline from NSCs to immature neurons. Once the cells become mature granule neurons, Notch signaling is turned on again (Figure 4). These data are presented in Results (subsection “Notch pathway elements are present in the SGZ NSC niche”) and the methodology in Materials and methods. Hes5 is shuttled between nucleus and cytoplasm to control its transcriptional activity (Gottle et al., 2015; Meliou et al., 2011); thus, cytoplasmic staining pattern is possible. Our staining protocol included usage of Tyramide Signal Amplification kit, which might have contributed to the punctuated staining pattern. To demonstrate the specificity of the signal, we have provided new images in the revised manuscript, which includes the GZ (Figure 4 – right panel). Due to specific localization of the signal along the GFAP+ process of NSCs in SGZ and absence of it from the GZ, we believe the staining pattern is genuine.

Finally, to demonstrate the reliability of Jag1 antibody, we stained sections from both wild type and i*Jag1*^fl/fl^ mice. Since CreERT2 is driven by *Lfng* promoter in i*Jag1*^fl/fl^ mice, the effect of *Jag1* deletion is expected to be restricted to SGZ and not other parts of the dentate gyrus. This gave us the advantage of a positive control in the conditional knockout tissue. Indeed, in i*Jag1*^fl/fl^ mice, we detected Jag1 staining in the hilar cells while the SGZ was mostly devoid of signal. In wild-type mice, we detected Jag1^+^ cell clusters in the SGZ and hilus, confirming the specificity of Jag1 staining (see Figure 8; scale bars=20µm). Finally, the specificity of the Dll1 antibody could not be verified since we have used Dll1 heterozygous constitutive knockout mice.

Author response image 1.Jag1 signal is mostly eliminated from SGZ in i*Jag1*^fl/fl^ mice.Immunostaining for Jag1 expression using anti-Jag1 antibody (Santa Cruz Biotechnology, Santa Cruz, CA) shows that the signal is localized to both SGZ and hilus in control (Ctrl) samples, whereas it is present mostly in hilus in the conditional knockout mice and not in the SGZ. Method is described in Materials and methods section. Scale bar=20µm.**DOI:**
http://dx.doi.org/10.7554/eLife.24660.014

*b) Perform similar Notch signaling analysis in their inducible Lfng knockout animals, so that they can directly compare Lfng wild type and knockout NSCs and their downstream progeny and demonstrate unequivocally that Notch signal is indeed affected in the Lfng knockout NSCs (and not other cell types).*

Please see the answer to reviewer 3, question 1a.

*c) Explain the discrepancies between Jag^+/-^ and inducible Jag1 knockout data.*

We can understand how the difference between the BrdU^+^ NSC ratios in the constitutive and conditional *Jag1* knockout mice might appear confusing. In constitutive heterozygous *Jag1* mice, we find no significant difference in BrdU^+^ NSCs ratio between mutant and wild-type mice, *but there is an upward trend*. In conditional homozygous *Jag1* mice, we find significantly increased BrdU^+^ NSCs ratio compared to control mice. This difference could be due to Jag1’s ability to either control cell cycle duration or the recruitment of NSCs from the quiescent population. As we stated in the original text, multiple observations considered together favor the first hypothesis and support the role of *Jag1* in regulation of the NSC cell cycle duration but not in recruitment of quiescent NSCs.

First, if *Jag1* had a role in the recruitment of NSCs, there should be significant difference between the BrdU^+^ NSC ratios of *Jag1*^-/+^ and wild-type mice, but this is not the case (Figure 6 middle panel, p=0.8393).

Second, increased recruitment should have a negative effect on NSC population in *Jag1*^-/+^ animals as in *Dll1*^-/+^, but there is no significant difference between wild-type and *Jag1*^-/+^ NSC populations (Figure 6).

Third, 6 month-old *Lfng*^-/+^ and double constitutive heterozygous knockout *Lfng*^-/+^; *Jag1*^-/+^ mice both have the same number of NSCs (Figure 6—figure supplement 1) and similar BrdU^+^ NSC ratio (Figure 6—figure supplement 1), suggesting that *Jag1* does not have an additional effect on the recruitment of NSCs. There was an increased NSC cell cycle re-entry after 1 week BrdU pulse (Figure 6—figure supplement 1), most likely contributed by indicating that in the double knockout, lack of *Lfng* and lack of *Jag1* have independent effects.

In sum, these data support the hypothesis that the difference between constitutive heterozygous (upward trend in BrdU^+^ ratio) and conditional homozygous (significant increase in BrdU^+^ ratio) *Jag1* knockout data derives from the accumulation of actively cycling NSCs rather than an increase in the recruitment of quiescent NSCs. The increase in NICD1 expression in individual mutant NSC clones in i*Jag1*^fl/fl^ mice two weeks after induction (Figure 6) supports this notion, in agreement with the reported conditional overexpression of NICD1, which increases cell cycle re-entry of NSCs (Breunig et al., 2007). To clarify our findings, we modified the text in the Results section (subsection “Notch ligands Jag1 and Dll1 preserve NSCs by opposing effects on their cell cycle”) and added to Discussion (sixth paragraph).

*d) Significantly revise and tone down their claims about Notch signaling and the action of Notch ligand Dll1 and Jag1 unless they can demonstrate knockout data that is more cell type specific.*

We have developed a reliable protocol for NICD1 staining to examine Notch signaling in NSCs and their progeny, and independently verified the reliability of the NICD1 staining in CBF:H2B-Venus mice, demonstrating that it overlaps more than 90% in the SGZ NSC niche. We have also shown that conditional removal of *Lfng* from NSCs results in decreased NICD1 intensity, whereas conditional removal of *Jag1* caused a significant increase in the NICD1 intensity in NSCs as well as in SGZ. Thus, in our opinion, we demonstrated that Notch signaling changes in *Lfng* and *Jag1* knockout mice specifically in NSCs. Nevertheless, we modified the text in the Results (subsections “Lfng preserves NSCs by controlling their cell cycle” and “Notch ligands Jag1 and Dll1 preserve NSCs by opposing effects on their cell cycle”) and Discussion to clarify our claims.

*2) The authors' interpretation for their BrdU, or CIdU/IdU phenotype in Lfng mutant and Dll1/Jag1 is confusing and difficult to understand. Some textual changes will be essential to make these data more accessible to the readers.*

*Specifically, their data suggest that in Lfng mutant, (i) label retaining NSCs decrease with time (which is not due to multiple rounds of cell cycle entry during the chase period since fewer NSCs enter S phase twice in Lfng mutant); (ii) label retaining astrocytes increase with time; (iii) more NSCs are proliferating at any given time; and (iv) NSCs numbers are reduced. If apoptosis status is not altered in Lfng mutants (see below), this data together with the lineage-tracing data suggest that upon loss of Lfng, more NSCs are in cycle at any given time, but there is a shift in the division outcomes with regard to what cells are produced. In wild type, many of these divisions generate ANPs and stem cells, while in Lfng knockout animals, more astrocytes are produced.*

*The authors' statements like "NSCs lacking Lfng mostly exited the cell cycle", "Lfng removal caused NSCs to exit cycling mode earlier", or "these data indicate that absence of Jag1 causes NSCs to be in cell cycle for a prolonged length of time, while absence of Dll1 causes them to exit cell cycle" are confusing, and send out the opposite message. "NSCs exit cell cycle" implies that "NSCs are withdrawn from the cell cycle and become quiescent but remain as NSCs". If this is the phenotype, they should have observed more BrdU label-retaining NSCs, not less. A more appropriate statement might be "NSCs differentiate into astrocytes". In all, the authors should modify all of these related statements to convey the results more truthfully and enhance the readability of their manuscript.*

We have edited the text to the best of our ability. We also performed the 30-day BrdU pulse-and-chase experiment in the *Dll1*^-/+^ mice and showed that more proliferating NSCs terminally differentiate into astrocytes (Figure 6). We present these data in Results and Discussion.

*3) The authors should also conduct apoptosis analysis to determine if changes in cell numbers are due to abnormal apoptosis, which is an independent mechanism that will influence interpretation.*

Due to the rapid clearance of apoptotic cells in the SGZ (Sierra et al., 2010), quantification of apoptosis is difficult to examine. We therefore performed apoptosis analysis with two methods to ensure our observations are valid: activated caspase-3 staining and ApopTag®, which detects double-strand DNA breaks. Both methods demonstrated a significant decrease in the number of apoptotic cells in both *iJag1*^fl/fl^ and *iLfng*^fl/fl^ animals compared to wild-type mice (Figure 5—figure supplement 1). In *iJag1*^fl/fl^, although increased production of ANPs was expected to result in a greater number of apoptotic cells, we actually observed significantly fewer apoptotic cells. In *iLfng*^fl/fl^ animals, Notch signaling is downregulated, so more apoptotic cells would be expected. However, we observed a significant decrease in the number of apoptotic cells (Figure 5—figure supplement 1), most likely due to the production of fewer ANPs in *iLfng*^fl/fl^ mice. We present these findings in Results, subsections “Lfng preserves NSCs by controlling their cell cycle” and “Notch ligands Jag1 and Dll1 preserve NSCs by opposing effects on their cell cycle”, and describe the methodology in Materials and methods.

*4) The authors should provide evidence suggesting that Lfng-CreER is not active without tamoxifen induction (i.e., no leakage issue that might confound the interpretation).*

We have added the control experiment (vehicle injection) for tamoxifen induction and no leakage was observed (Figure 3—figure supplement 1).